# ReST-MCTS*: LLM Self-Training via Process Reward Guided Tree Search

**Dan Zhang**[1*†], **Sining Zhoubian**[1*], **Ziniu Hu**[2*], **Yisong Yue**[2], **Yuxiao Dong**[1], **Jie Tang**[1]

[1]The Knowledge Engineering Group (KEG), Tsinghua University;
[2]California Institute of Technology
{zd21,zbsn21}@mails.tsinghua.edu.cn
https://rest-mcts.github.io/

## Abstract

Recent methodologies in LLM self-training mostly rely on LLM generating responses and filtering those with correct output answers as training data. This approach often yields a low-quality fine-tuning training set (e.g., incorrect plans or intermediate reasoning). In this paper, we develop a reinforced self-training approach, called **ReST-MCTS***, based on integrating process reward guidance with tree search MCTS* for collecting higher-quality reasoning traces as well as per-step value to train policy and reward models. ReST-MCTS* circumvents the per-step manual annotation typically used to train process rewards by tree-search-based reinforcement learning: Given oracle final correct answers, ReST-MCTS* is able to infer the correct process rewards by estimating the probability this step can help lead to the correct answer. These inferred rewards serve dual purposes: they act as value targets for further refining the process reward model and also facilitate the selection of high-quality traces for policy model self-training. We first show that the tree-search policy in ReST-MCTS* achieves higher accuracy compared with prior LLM reasoning baselines such as Best-of-N and Tree-of-Thought, within the same search budget. We then show that by using traces searched by this tree-search policy as training data, we can continuously enhance the three language models for multiple iterations, and outperform other self-training algorithms such as ReST[EM] and Self-Rewarding LM. We release all code at https://github.com/THUDM/ReST-MCTS.

## 1 Introduction

Large Language Models (LLMs) are mostly trained on human-generated data. But as we approach the point where most available high-quality human-produced text on the web has been crawled and used for LLM training [1], the research focus has shifted towards using LLM-generated content to conduct self-training [2; 3; 4; 5; 6; 7]. Similar to most Reinforcement Learning (RL) problems, LLM self-training requires a reward signal. Most existing reinforced self-improvement approaches (e.g., STaR [4], RFT [5], ReST[EM] [6], V-STaR [7]) assume to have access to a ground-truth reward model (labels from supervised dataset, or a pre-trained reward model). These approaches use an LLM to generate multiple samples for each question, and assume the one that leads to high reward (correct solution) is the high-quality sample, and later train on these samples (hence self-training). Such procedures can be effective in improving LLM performance, in some cases solving reasoning tasks that the base LLM cannot otherwise solve [8; 9; 10].

However, a key limitation of the above procedure is that even if a reasoning trace results in a correct solution, it does not necessarily imply that the entire trace is accurate. LLMs often generate

---

[*]Equal contribution.
[†]Work done while DZ visited at Caltech.

38th Conference on Neural Information Processing Systems (NeurIPS 2024).

Table 1: Key differences between existing self-improvement methods and our approach. Train refers to whether to train a reward model.

| Method | Reasoning Policy | Reward Guidance | | Train |
|---|---|---|---|---|
| | Method | | Value Label | |
| STaR [4] | CoT+Reflexion | **Final outcome reward** annotated by ground-truth answer | | ✗ |
| ReST^EM [6] / RFT [5] / RPO [11] | CoT | | | ✗ |
| Verify Step-by-Step [1] | Best-of-N | **Per-step process reward** annotated by human | | ✓ |
| MATH-SHEPHERD [12] / pDPO [13] | Best-of-N | **Per-step process reward** inferred from random rollout | | ✓ |
| TS-LLM [14] | MCTS | **Per-step process reward** inferred from TD-$\lambda$ [15] | | ✓ |
| V-STaR [7] | CoT | **Final outcome reward** generated by multi-iteration LLMs | | ✓ |
| Self-Rewarding [16] | CoT | **Final outcome reward** generated and judged by LLMs | | ✗ |
| **ReST-MCTS\*** (Ours) | MCTS\* | **Per-step process reward** inferred from tree search (MCTS\*) | | Multi-Iter |

wrong or useless intermediate reasoning steps, while still finding the correct solution by chance [17]. Consequently, a self-training dataset can often contain many false positives — intermediate reasoning traces or plans are incorrect, but the final output is correct — which limits the final performance of LLM fine-tuning for complex reasoning tasks [18; 19]. One way to tackle this issue is to use a value function or reward model to verify reasoning traces for correctness (which then serves as a learning signal for self-training) [1; 12]. However, training a reliable reward model to verify every step in a reasoning trace generally depends on dense human-generated annotations (per reasoning step) [1], which does not scale well. Our research aims to address this gap by developing a novel approach that automates the acquisition of reliable reasoning traces while effectively utilizing reward signals for verification purposes. Our key research question is: **How can we automatically acquire high-quality reasoning traces and effectively process reward signals for verification and LLM self-training?**

In this paper, we propose ReST-MCTS\*, a framework for training LLMs using model-based RL training. Our proposed approach utilizes a modified Monte Carlo Tree Search (MCTS) algorithm as the reasoning policy, denoted MCTS\*, guided by a trained per-step process reward (value) model. A key aspect of our method is being able to automatically generate per-step labels for training per-step reward models, by performing a sufficient number of rollouts. This labeling process effectively filters out the subset of samples with the highest quality, without requiring additional human intervention. Table 1 summarizes the key distinctions between our approach and previous approaches. We validate experimentally that ReST-MCTS\* outperforms prior work in discovering good reasoning traces, such as Self-Consistency (SC) and Best-of-N (BoN) under the same search budget on the SciBench [20] and MATH [21] benchmarks, which consequently leads to improved self-training.

To summarize, our contributions are:

- We propose ReST-MCTS\*, a self-training approach that generates process rewards searched by MCTS\*. A key step is to automatically annotate the process reward of each intermediate node via sufficient times of rollouts, using MCTS\*. We validate multiple reasoning benchmarks and find that ReST-MCTS\* outperforms existing self-training approaches (e.g., ReST^EM and Self-Rewarding) as shown in Table 2 and reasoning policies (e.g., CoT and ToT) as shown in Table 4.

- The reward generator in ReST-MCTS\* leads to a higher-quality process reward model compared to previous process reward generation techniques, e.g., MATH-SHEPHERD, as shown in Table 3.

- Given the same search budget, the search algorithm (MCTS\*) in ReST-MCTS\* achieves higher accuracy than Self-Consistency and Best-of-N, as shown in Figure 2.

## 2 Background on Reasoning & Self-Training

We follow the standard setup in LLM-based reasoning. We start with a policy, denoted by $\pi$, that is instantiated using a base LLM. Given an input problem $Q$, in the simplest case, $\pi$ can generate an output sequence, or trace, of reasoning steps $(s_1, s_2, \cdots, s_K) \sim \pi(\cdot|Q)$ by autoregressively predicting the next token. For simplicity, we assume a reasoning step comprises a single sentence (which itself comprises multiple tokens). We also assume the last output $s_K$ is the final step. LLMs can also be prompted or conditioned to bias the generation along certain traces. For a prompt $c$, we can write the policy as $\pi(\cdot|Q, c)$. This idea was most famously used in chain-of-thought (CoT) [22].

**Self-Consistency (SC).** Self-Consistency [23] samples multiple reasoning traces from $\pi$ and chooses the final answer that appears most frequently.

**Tree-Search & Value Function.** Another idea is to use tree-structured reasoning traces [24; 14], that branch from intermediate reasoning steps. One key issue in using a so-called tree-search reasoning algorithm is the need to have a value function to guide the otherwise combinatorially large search process [14]. Two common value functions include Outcome Reward Models (ORMs) [25], which are trained only on the correctness of the final answer, and Process Reward Models (PRMs) [1], which are trained on the correctness of each reasoning step. We assume $r_{s_k}$ is the PRM's output sigmoid score at $k$-th step. Our ReST-MCTS* approach uses tree-search to automatically learn a good PRM.

**Best-of-N.** As an alternative to Self-Consistency, one can also use a learned value function (PRM or ORM) to select the reasoning trace with the highest value [1].

**Self-Training.** At a high level, there are two steps to self-training [6; 12]. The first step is generation, where we sample multiple reasoning traces using $\pi$ (in our case, tree-structured traces). The second step is improvement, where a learning signal is constructed on the reasoning traces, which is then used to fine-tune $\pi$. The process can repeat for multiple iterations.

**Limitation of Prior Works.** The main challenge in doing reliable self-training is the construction of a useful learning signal. Ideally, one would want a dense learning signal on the correctness of every intermediate reasoning step, which is given by a PRM. Otherwise, with sparse learning signals, one suffers from a credit assignment similar to that in reinforcement learning. Historically, the main challenge with learning a PRM is the lack of supervised annotations per reasoning step. This is the principal challenge that our ReST-MCTS* approach seeks to overcome. We describe detailed preliminaries in Appendix A.

# 3 The ReST-MCTS* Method

Our approach, ReST-MCTS*, is outlined in Figure 1 and developed using four main components.

- **MCTS*** which performs a tree search with sufficient rollout time under the guidance of the PRM.
- **Process Reward Model** (PRM) which evaluates any partial solution's quality and guides MCTS*.
- **Policy Model** which generates multiple intermediate reasoning steps for each question.
- **LLM Self-Training**, which uses MCTS* to collect reasoning traces, trains policy model on positive samples, and trains process reward model on all generated traces.

## 3.1 Search-based Reasoning Policy for LLM

**Value $v_k$ for a Partial Solution.** The value (process) reward $v_k$ of the partial solution $p_k = [s_1, s_2, \cdots, s_k]$ should satisfy the following basic qualities:

- Limited range: $v_k$ is constrained within a specific range. This restriction ensures that the values of $v_k$ are bounded and do not exceed a certain limit.
- Reflecting probability of correctness: $v_k$ reflects the probability that a partial solution is a complete and correct answer. Higher values of $v_k$ indicate better quality or a higher likelihood of being closer to a correct answer.
- Reflecting correctness and contribution of solution steps: $v_k$ incorporates both the correctness and contribution of each solution step. When starting from a partial solution, a correct next step should result in a higher $v_k$ compared to false ones. Additionally, a step that makes more correct deductions toward the final answer should lead to a higher $v_k$ value. This property ensures that $v_k$ captures the incremental progress made towards the correct solution and rewards steps that contribute to the overall correctness of the solution.

**Reasoning Distance $m_k$ for a Partial Solution.** To estimate the progress of a solution step, we define the reasoning distance $m_k$ of $p_k$ as the minimum reasoning steps a policy model requires to reach the correct answer, starting from $p_k$. Reasoning distance reflects the progress made as well as the difficulty for a policy to figure out a correct answer based on current steps, thus it can be further

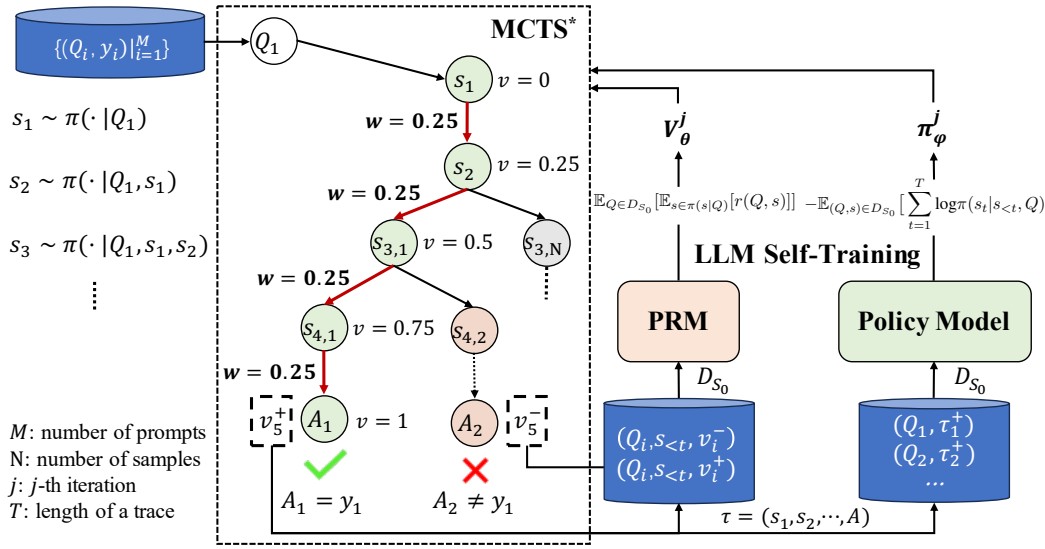

Figure 1: The left part presents the process of inferring process rewards and how we conduct process reward guide tree-search. The right part denotes the self-training of both the process reward model and the policy model.

used to evaluate the quality of $p_k$. However, we point out that $m_k$ can not be directly calculated. It is more like a hidden variable that can be estimated by performing simulations or trace sampling starting from $p_k$ and finding the actual minimum steps used to discover the correct answer.

**Weighted Reward $w_{s_k}$ for a Single Step.** Based on the desired qualities for evaluating partial solutions, we introduce the concept of a weighted reward to reflect the quality of the current step $s_k$, denoted as $w_{s_k}$. Based on the common PRM reward $r_{s_k}$, $w_{s_k}$ further incorporates the reasoning distance $m_k$ as a weight factor, reflecting the incremental progress $s_k$ makes.

**Representations for Quality Value and Weighted Reward.** To determine the quality value $v_k$ of a partial solution at step $k$, we incorporate the previous quality value and the weighted reward of the current step. By considering the previous quality value, we account for the cumulative progress and correctness achieved up to the preceding step. Therefore, the $v_k$ can be iteratively updated as:

$$v_k = \begin{cases} 0, & k = 0 \\ max(v_{k-1} + w_{s_k}, 0), & else \end{cases} \tag{1}$$

The weighted reward $w_{s_k}$ of the current step provides a measure of the quality and contribution of that specific step towards the overall solution. Based on $m_k$ (where $m_k = K - k$ and $K$ is the total number of reasoning steps of a solution $s$), previous quality value $v_{k-1}$, and $r_{s_k}$ in MATH-SHEPHERD [12], we can update the definition of the weighted reward $w_{s_k}$ iteratively as follows:

$$w_{s_k} = \frac{1 - v_{k-1}}{m_k + 1}(1 - 2r_{s_k}), \ \ k = 1, 2, \cdots \tag{2}$$

As $k$ increases, $m_k$ decreases, indicating that fewer reasoning steps are needed to reach the correct answer. This leads to a higher weight placed on the weighted reward of the current step. We can also derive that $w_{s_k}$ and $v_k$ satisfy the expected boundedness shown in the theorem below.

**Theorem 1** (Boundedness of $w_{s_k}$ and $v_k$). *If $r_{s_k}$ is a sigmoid score ranged between $[0, 1]$, then $w_{s_k}$ and $v_k$ defined as above satisfy following boundedness: $w_{s_k} \leq 1 - v_{k-1}$, $v_k \in [0, 1]$.*

*Derivation.* Please refer to the detailed derivation in Appendix B.1.

Therefore, we can conclude that $w_{s_k}$ and $v_k$ has following properties that match our expectations:

> **Observation 1.** *If a reasoning route starting from $p_k$ requires more steps to get to the correct answer, then the single-step weighted reward $w_{s_k}$ is lower.*

> **Observation 2.** $w_{s_k}$ *decreases as the PRM's predicted sigmoid score $r_{s_k}$ rises. Thus, $w_{s_k}$ has a positive correlation with the PRM's prediction of a step's correctness.*

> **Observation 3.** $v_k \rightarrow 1 \iff r_{s_k} \rightarrow 0,\ m_k = 0$, *i.e. $v_k$ converges to upper bound 1 only when $s_k$ reaches the correct answer.*

Based on the features of $v_k$ and $w_{s_k}$, we can directly predict the quality value of partial solutions and guide search once we have a precise PRM and accurate prediction of $m_k$. In our approach, instead of separately training models to predict $r_{s_k}$ and $m_k$, we simply train a process reward model $V_\theta$ to predict $v_k$, serving as a variant of common PRM. With reward incorporated in the calculation of $v_k$, there is no need to separately train a reward model, saving considerable effort for answer selection.

**Process Reward Model Guided Tree Search MCTS\*.** Tree search methods like [24] and [26] require a value function and outcome reward model $r_\phi$ to prune branches, evaluate final solutions and backup value. However, using ORM to evaluate final solutions and backpropagate means every search trace must be completely generated, which is costly and inefficient. Recent work [14] suggests using a learned LLM value function in MCTS so the backup process can happen in the intermediate step, without the need for complete generations. Their work greatly improves search efficiency but still relies on an ORM to select the final answer. Drawing inspiration from these works, we further propose a new variant of MCTS, namely **MCTS\***, which uses quality value $v_k$ as a value target for a trained LLM-based process reward model and guidance for MCTS as well.

Given the above properties, we can directly use the process reward model $V_\theta$ to evaluate the quality of any partial solution, select, and backpropagate in intermediate nodes. Aside from the use of quality value, we also incorporate a special Monte Carlo rollout method and self-critic mechanism to enhance efficiency and precision, which are explained detailedly in Appendix C.1. We express MCTS\* as an algorithm that comprises four main stages in each iteration, namely node selection, thought expansion, greedy MC rollout, and value backpropagation. Similar to common MCTS settings, the algorithm runs on a search tree $T_q$ for each single science reasoning question $q$. Every tree node $C$ represents a series of thoughts or steps, where a partial solution $p_C$, number of visits $n_C$, and corresponding quality value $v_C$ are recorded. For simplicity, we denote each node as a tuple $C = (p_C, n_C, v_C)$. An overall pseudo-code for MCTS\* is presented in Algorithm 2.

## 3.2 Self-Training Pipeline

As shown in Figure 1, based on the proposed tree search algorithm MCTS\*, we perform self-improvement on the reasoning policy and process reward model. After initialization of the policy $\pi$ and process reward model $V_\theta$, we iteratively employ them and utilize the search tree $T_q$ generated in the process to generate high-quality solutions for specific science or math questions and conduct a self-improvement process, called ReST-MCTS\*. Our work draws inspiration from the MuZero [20] framework and applies it to the training of LLMs which we term "MuZero-style learning of LLMs".

**Instruction Generation.** In this stage, initialization starts from an original dataset $D_0$ for the training process reward model $V_\theta$.

• **Collect process reward for process reward model.** The extraction of new value data is relatively more complex, we derive the target quality value of partial solutions of every tree node near a correct reasoning path on the pruned search tree $T_q'$. We first calculate $m_k$ for every tree node $C$ that is on at least one correct reasoning trace (including the root) according to its minimum reasoning steps required to get to a correct answer in $T_q'$. Then, we use the hard estimation in Eq. (11) in [12] to calculate $r_{s_k}$, i.e. $r_{s_k} = 1 - r_{s_k}^{\text{HE}}$, which means a reasoning step is considered correct if it can reach a correct answer in $T_q'$. Using $m_k$ and $r_{s_k}$, we are able to derive the value of the partial solution of every node on or near one correct reasoning trace. For each node $C$ (with partial solution $p_C = [s_1, s_2, \cdots, s_{k-1}]$) on at least one correct trace and a relevant forward step $s_k$, we can derive the value $v_k$ using Eq. (1) and weighted reward $w_k$ using Eq. (2), with $m_k$ set to the same as $m_{k-1}$ if $r_{s_k}^{\text{HE}} = 0$ in Eq. (11). A concrete and detailed example of this inferring process is shown in Figure 3. We update all these rewards and values starting from the root and collect all $(Q, p, v)$ pairs to form $D_{V_i}$ in $i$-th iteration, which is used for training a process reward model in the next iteration.

• **Collect reasoning traces for policy model.** As shown in Figure 4, the search process produces a search tree $T_q$, consisting of multiple reasoning traces. We first prune all the unfinished branches (branches that do not reach a final answer). Then we verify other traces' final answers acquired in the tree search according to their correctness through simple string matching or LLM judging and select the correct solutions. These verified reasoning traces, as $D_{G_i(A_j=a^*)|_{j=1}^N}$ (where $N$ is the number of sampling solutions, $A_j$ is the $j$-th solution, and $a^*$ is the final correct answer) in $i$-th iteration, are then used for extracting new training data for policy self-improvement. This process is followed by Eq. (13) ($i \geq 1$) to execute the policy self-training.

**Mutual Self-training for Process Reward Model and Policy Model.** Compared to previous work like ReST$^{EM}$ [6], which only concerns self-training for the policy and demonstrates that the policy can improve by iteratively generating new traces and learning from the high-reward ones generated by itself, our work simultaneously improves the process reward model and policy model self-training. With the process reward model's training set $D_{V_0}$ initialized and new problem set $D_G$ given, we can start the iterative self-training process upon $V_\theta$ and $\pi$. We use $\pi$ to perform MCTS$^*$ and generate solutions for $D_G$, with implement details illustrated in Section 3.1. In the $i$-th ($i = 1, 2, \cdots$) iteration, we train $V_\theta$ with $D_{V_{i-1}}$ to obtain $V_i$ and train policy model $\pi_{S_{i-1}}$ on $D_{G_i}$ to generate new generator $\pi_{S_i}$. At the same time, $D_{G_i}$ drives the update of $V_i$ to $V_{i+1}$. We present iterative self-training that the process reward model and policy model complement each other in Algorithm 1.

---

**Algorithm 1:** Mutual self-training ReST-MCTS$^*$ for value model and policy model.

---

**Input:** base LLM $\pi$, original dataset for policy model $D_{S_0}$, original dataset for value model $D_0$, new problem set $D_G$, number of solutions $N$, $j$-th solution $A_j$, correct solution $a^*$, value model $V_\theta$, weighted value function $w$, quality value function $v$, number of iterations $T$.

1:   $\pi_{S_0} \leftarrow$ SFT$(\pi, D_{S_0})$ // `fine-tune generator`
2:   $D_{V_0} \leftarrow$ generate_value_data$(D_0, w, v)$ // `initialize train set for value model`
3:   $V_0 \leftarrow$ train_value_model$(V_\theta, D_{V_0})$ // `initialize value model`
4:   **for** $i = 1$ to $T$ **do**
5:      $D_{G_i} \leftarrow$ generate_policy_data$(\pi_{S_{i-1}}, V_{i-1}$ guided MCTS$^*, D_G, N)$ // `generate`
       `synthetic data for policy model`
6:      **for** $j = 1$ to $N$ **do**
7:        $D_{G_i(A_j=a^*)} \leftarrow$ label_correctness$(D_{G_i})$ // `match and select correct solutions`
8:      **end for**
9:      $\pi_{S_i} \leftarrow$ SFT$(\pi_{S_{i-1}}, D_{G_i(A_j=a^*)|_{j=1}^N})$ // `self-training policy model`
10:     $D_{V_i} \leftarrow$ extract_value_data$(D_{G_i})$ // `collect process reward and extract value data`
11:     $V_i \leftarrow$ train_value_model$(V_{i-1}, D_{V_i})$ // `self-training value model`
12: **end for**
**Output:** $\pi_{S_T}, V_T$

---

## 4 Experiments

We validate ReST-MCTS$^*$ from three perspectives:

• **Self-Training approaches** which use generated samples and evaluated for multiple iterations, such as ReST$^{EM}$ and Self-Rewarding, on in-distribution and out-of-distribution benchmarks under three LLM backbones, as shown in Table 2. ReST-MCTS$^*$ outperforms existing approaches in each iteration and continuously self-improves with the data it generates.

• **Process reward models** which are compared with the state-of-the-art techniques, such as MATH-SHEPHERD (MS) and SC + MS on GSM8K and MATH500, as shown in Table 3. Results indicate that the ReST-MCTS$^*$ learns a good PRM and our reward model implements higher accuracy.

• **Tree-Search policy** which are compared on college-level scientific reasoning benchmark under three LLMs, such as CoT and ToT, as shown in Table 4. We also evaluated under the same search budget on MATH and SciBench, such as SC and Best-of-N, as shown in Figure 2. Results show the ReST-MCTS$^*$ significantly outperforms other baselines despite insufficient budget.

Table 2: Primary results by training both policy and value model for multiple iterations. For each backbone, different self-training approaches are conducted separately. This means each approach has its own generated train data and corresponding reward (value) model. Our evaluation is zero-shot only, the few-shot baseline only serves as a comparison.

| Model | Self-Training Methods | MATH | GPQA_Diamond | CEval-Hard | Ave. |
|---|---|---|---|---|---|
| | 0th iteration (zero-shot) | 20.76 | 27.27 | 26.32 | 24.78 |
| | 0th iteration (few-shot) | 30.00 | 31.31 | 25.66 | 28.99 |
| | (Below are fine-tuned from model of previous iteration with self-generated traces) | | | | |
| LLaMA-3-8B-Instruct | w/ ReST$^{EM}$ (1st iteration) | 30.84 | 26.77 | 21.05 | 26.22 |
| | w/ Self-Rewarding (1st iteration) | 30.34 | 26.26 | 25.66 | 27.42 |
| | w/ **ReST-MCTS*** (**1st iteration**) | 31.42 | 24.24 | 26.97 | 27.55 |
| | w/ ReST$^{EM}$ (2nd iteration) | 33.52 | 25.25 | 21.71 | 26.83 |
| | w/ Self-Rewarding (2nd iteration) | 33.89 | 26.26 | 23.03 | 27.73 |
| | w/ **ReST-MCTS*** (**2nd iteration**) | 34.28 | 27.78 | 25.00 | **29.02** |
| | 0th iteration (zero-shot) | 29.34 | 27.78 | 9.87 | 22.33 |
| | 0th iteration (few-shot) | 28.28 | 29.29 | 9.21 | 22.26 |
| | (Below are fine-tuned from model of previous iteration with self-generated traces) | | | | |
| Mistral-7B: MetaMATH | w/ ReST$^{EM}$ (1st iteration) | 23.84 | 26.26 | 20.39 | 23.50 |
| | w/ Self-Rewarding (1st iteration) | 25.70 | 27.78 | 19.74 | 24.40 |
| | w/ **ReST-MCTS*** (**1st iteration**) | 31.06 | 26.26 | 17.11 | 24.81 |
| | w/ ReST$^{EM}$ (2nd iteration) | 23.86 | 26.26 | 22.37 | 24.16 |
| | w/ Self-Rewarding (2nd iteration) | 23.90 | 26.77 | 25.00 | 25.22 |
| | w/ **ReST-MCTS*** (**2nd iteration**) | 24.40 | 28.79 | 26.32 | **26.50** |
| | 0th iteration | 25.18 | 23.74 | 51.97 | 33.63 |
| | (Below are fine-tuned from model of previous iteration with self-generated traces) | | | | |
| SciGLM-6B | w/ ReST$^{EM}$ (1st iteration) | 22.72 | 24.75 | 51.32 | 32.93 |
| | w/ Self-Rewarding (1st iteration) | 22.50 | 26.26 | 47.37 | 32.04 |
| | w/ **ReST-MCTS*** (**1st iteration**) | 24.86 | 25.25 | 51.32 | 33.81 |
| | w/ ReST$^{EM}$ (2nd iteration) | 25.86 | 25.25 | 48.68 | 33.27 |
| | w/ Self-Rewarding (2nd iteration) | 23.86 | 28.79 | 48.03 | 33.56 |
| | w/ **ReST-MCTS*** (**2nd iteration**) | 23.90 | 31.82 | 51.97 | **35.90** |

## 4.1 Initialization of Value Model

To obtain accurate feedback from the environment, we build the value model's initial train set $D_{V_0}$ from a set of selected science or math questions $D_0$ using process reward (value) inference, with no human labeling process required. Then, we finetune the ChatGLM3-6B [27; 28] and Mistral-7B [29] model on this dataset, respectively, obtaining initial value models that, as variants of PRM, guide the LLM tree search for higher-quality solutions upon both math and science questions.

**Fine-grained dataset for science and math.** Aiming to gather value train data for science, we integrate questions of a lean science dataset $D_{sci}$ within SciInstruct [10] into $D_0$. This dataset consists of 11,554 questions, where each question is paired with a correct step-by-step solution. For each question $q^{(i)}(i = 1, 2, \cdots, N)$ and corresponding solution $s^{(i)} = s^{(i)}_{1,2,\cdots,K_i}$ in $D_{sci}$, we extract all partial solutions to form samples $d^{(i)}_k = [q^{(i)}, s^{(i)}_{1,2,\cdots,k}(p^{(i)}_k)](k = 1, 2, \cdots, K_i)$. To make the value model distinguish false steps, we also employ a LLM policy (ChatGLM2) that is basically incompetent for reasoning tasks of this difficulty to generate single steps $s^{(i)'}_{k+1}$ given $q^{(i)}$ and $p^{(i)}_k$, obtaining new partial solutions $p^{(i)'}_{k+1} = [s^{(i)}_{1,2,\cdots,k}, s^{(i)'}_{k+1}]$ and new samples $d^{(i)}_{k,j} = [q^{(i)}, s^{(i)}_{1,2,\cdots,k}, s^{(i)'}_{k+1,j}](j = 1, 2, 3)$. For simplicity, the generated steps are regarded as incorrect. In total, we collect 473.4k samples for training the initial value model. Afterward, we derive target quality values for all samples $d^{(i)}_{k,j}$ and $d^{(i)}_k$ and use them to construct $D_{V_0}$, which is illustrated in Appendix B.1. We adopt an alternative method to generate value train data for math, as shown in Appendix B.1.

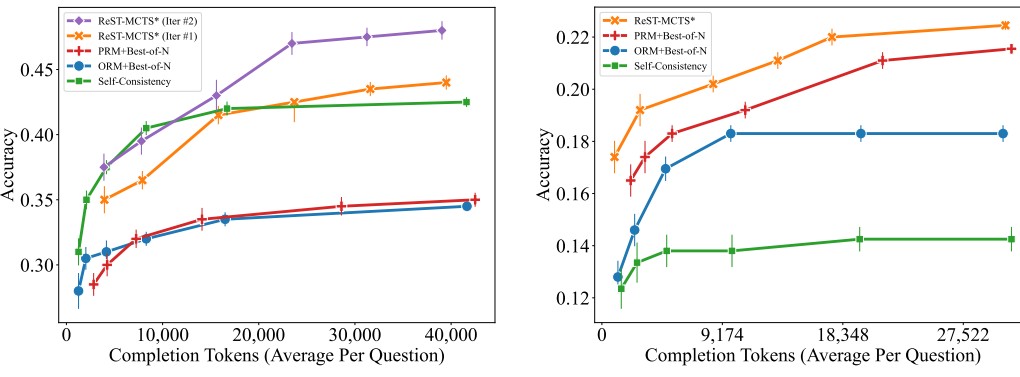

(a) Self-training of value model on MATH.    (b) Comparison of value model on SciBench.

Figure 2: Accuracy of different searches on MATH and SciBench with varied sampling budget.

Table 3: Accuracy of different verifiers on GSM8K test set and MATH500. SC: Self-Consistency, MS: MATH-SHEPHERD. Verification is based on 256 outputs.

| Models | Dataset | SC | ORM | SC+ORM | MS | SC + MS | SC + ReST-MCTS* (Value) |
|---|---|---|---|---|---|---|---|
| Mistral-7B: MetaMATH | GSM8K | 83.9 | 86.2 | 86.6 | 87.1 | 86.3 | **87.5** |
|  | MATH500 | 35.1 | 36.4 | 38.0 | 37.3 | 38.3 | **39.0** |

## 4.2 Evaluating Self-Improvement of ReST-MCTS*

In order to thoroughly examine the influence of ReST-MCTS* self-training on varied backbones, we execute 2 iterations of self-training and compare two representative self-training approaches, ReST$^{EM}$, which compares outcome reward with ground-truth answer, and Self-Rewarding, which judges outcome reward by LLMs, upon 3 different base models, namely LLaMA-3-8B-Instruct [30], Mistral-7B: MetaMATH [29; 31] and SciGLM-6B [10]. Primary results are shown in Table 2. Concerning the dataset for sample generation, since we are primarily interested in the continuous improvement ability of ReST-MCTS* in a specific domain, we mainly include math questions in the dataset. For simplicity, we use the same dataset $D_G$ in each iteration. It involves questions selected from a train set of well-known benchmarks including MATH, GSM8K, and TheoremQA [32]. With the policy and value model trained simultaneously on samples generated from $D_G$, we observe that our self-training paradigm enables continuous enhancement of the capabilities of both models on in-distribution and out-of-distribution benchmarks, regardless of which backbone is used.

• **Iterative performance improvement on policy model.** Previous LLM self-training approaches mostly rely on the generating responses of LLM and assume each question with the correct solution is a high-quality sample while the intermediate reasoning steps are wrong or useless in many cases. Therefore, we compare the ReST-MCTS* with recent self-training paradigms by generating new samples under different reward (value) supervision strategies. For ReST$^{EM}$ and Self-Rewarding, the default sampling strategy is generating CoT data, with generated data refined according to ground truth or reward provided by the policy, respectively. In comparison, ReST-MCTS* generates data samples via MCTS*, with data refined referring to quality value and ground truth. The results in Table 2 show that all three backbones can be continuously self-improved by data generated by itself, using ReST-MCTS* as a paradigm. ReST-MCTS* significantly outperforms previous self-training methods ReST$^{EM}$ and Self-Rewarding basically in each iteration. This means the ReST-MCTS* can screen out self-generated data of higher quality for better self-improvement.

• **Iterative performance improvement on reward model.** We also compare how our iterative trained policy and value model can improve the overall search results under the same token usage on the test set of MATH [33]. See implementation details in Appendix E.3. We show results in Figure 2 (a), where ReST-MCTS* (Iter #1) greatly outperforms most baselines but does not completely surpass Self-Consistency. In comparison, after more iterations of self-training, verification based on the enhanced value model basically outperforms Self-Consistency on every point, achieving the highest accuracy of $48.5\%$ that significantly exceeds the $42.5\%$ of Self-Consistency. This indicates the effectiveness of our self-training pipeline.

Table 4: Overall performance comparison with representative models on SciBench.

| Models | Subject | Chemistry | | | | Physics | | | Math | | | All |
|---|---|---|---|---|---|---|---|---|---|---|---|---|
| | Method | atkins | chemmc | quan | matter | fund | class | thermo | diff | stat | calc | Ave. |
| GLM4 | CoT | 11.21 | 23.07 | 8.82 | 4.08 | 19.44 | 2.12 | **7.46** | 10.00 | 12.00 | 28.57 | 12.68 |
| | ToT | 11.21 | 23.07 | 8.82 | **12.24** | **22.22** | **6.38** | 5.97 | **12.00** | 25.33 | **30.95** | 15.82 |
| | ReST-MCTS* | **13.08** | **28.20** | **14.70** | 8.16 | **22.22** | 4.25 | **7.46** | **12.00** | **26.66** | **30.95** | **16.77** |
| GPT-3.5-turbo | CoT | 5.60 | 7.69 | 5.88 | **6.12** | 6.94 | 2.12 | **2.98** | 4.00 | 16.00 | **11.90** | 6.92 |
| | ToT | **8.41** | **12.82** | **11.76** | **6.12** | **11.11** | 0.00 | 0.00 | **10.00** | 18.66 | 9.52 | 8.44 |
| | ReST-MCTS* | 5.60 | **12.82** | **11.76** | **6.12** | 6.94 | **8.51** | **2.98** | **10.00** | **24.00** | **11.90** | **10.06** |
| LLaMA2-13B-Chat | CoT | **2.80** | 2.56 | **2.94** | 2.04 | 2.77 | **2.12** | 0.00 | 2.00 | 2.66 | **2.38** | 2.23 |
| | ToT | 0.93 | **5.12** | **2.94** | **4.08** | 2.77 | 0.00 | **1.49** | 0.00 | 4.00 | **2.38** | 2.37 |
| | ReST-MCTS* | 0.93 | **5.12** | **2.94** | 2.04 | **4.16** | **2.12** | 0.00 | **4.00** | **5.53** | **2.38** | **2.90** |

## 4.3 Evaluating Reward Guidance and Reasoning Policy of ReST-MCTS*

Our main hypothesis in this paper is that a better search policy getting higher-quality traces can improve self-training. In this section, we mainly focus on whether our process reward guided MCTS* can gain improvement to get better samples over different reasoning tasks. We first evaluate the effectiveness of the value model itself standalone in Table 3 and then evaluate the performance of different reasoning policies in Table 4.

**Performance Comparison of Various Verification Models.** As [1] suggested, different value models or reward models vary in accuracy and fineness. We perform tests on the questions of the GSM8K and MATH500 using multiple reward models and verification methods. It is worth noting that we include the same experiment settings of MATH-SHEPHERD (MS) [12] as a comparison since it also adopts an automatic train data generation method for reward models. For SC+ReST-MCTS*, we utilize the same CoT-based sampling strategy as MS, except that SC is performed according to our own value model's output rather than the reward model of MS, which makes this a direct comparison of different reward model training approaches. We record the model accuracy of Mistral-7B: MetaMATH on the selected test set, which is as shown in Table 3. Results indicate that compared to MS and SC+MS, SC+ReST-MCTS* (Value) exhibits higher improvement in solution accuracy on both GSM8K and MATH. This confirms the effectiveness of our value model, further indicating that our definitions of quality value and weighted reward are valid or possibly even better.

**Performance Comparison under the Same Search Budget.** Though the MCTS-based search methods demonstrate significant improvement in model performance, they often require a considerable amount of token input and completion, which makes it quite costly in some circumstances. Therefore, we conduct more experiments to investigate the relationship between search token budget and model performance on science questions selected from SciBench comparing ReST-MCTS* and the same baselines employed for MATH, which are elaborated in Appendix E.3. Since our self-training procedure is primarily conducted on math data, so we do not consider the effects of self-training in this case. However, we point out that this can still be further investigated as a study of transfer learning for self-training paradigms. Figure 2 (b) shows the accuracy of different approaches on SciBench when the completion budget changes. Results indicate that the ReST-MCTS* greatly outperforms other baselines despite insufficient budget. We notice that although CoT-based methods can improve greatly by increasing the sample budget, they tend to quickly converge to a limited accuracy, which is not as satisfying as the ReST-MCTS*.

**Performance Comparison of Different Reasoning Policies on Benchmarks.** To evaluate the effectiveness of ReST-MCTS*, we perform benchmark experiments on SciBench [34] in Tabel 4 and SciEval [35] in Table 8. All benchmark setups are illustrated in Appendix E.2. For the backbone of models, large-scale models GLM4 and GPT-3.5-turbo (both API), as well as a small-scale model LLaMA2-13B-Chat are included. As shown in Table 4, with the experiment repeated for 2 times, we report the average accuracy scores (%) of 3 methods on 10 subjects. Concerning overall accuracy, the ReST-MCTS* outperforms other baselines for all 3 models, with GLM4 improved over 4.0% and GPT-3.5-turbo over 3.1%. On specific subjects such as chemmc, quan, and stat, the ReST-MCTS* achieves significant improvement over 5.0%, indicating its great potential in discovering accurate solutions. Besides, we notice that our ToT baseline also performs well on many subjects, sometimes even surpassing ReST-MCTS*. This reflects that our value model can provide appropriate guidance for tree-search-based methods. We also discovered that for LLaMA2-13B-Chat, the improvement is not very prominent. This reveals that small-scale policies may face difficulties when adopting complex tree search approaches since their capability for step-wise inference is relatively low.

# 5 Related Work

## 5.1 Large Language Model Training

Large Language Models (LLMs) [36; 37; 38] have emerged as a notable success in various natural language tasks. Recent studies focus on improving the reasoning capabilities of LLMs, including collecting high-quality or larger domain-specific data [39; 40; 41; 42; 10; 43], designing elaborate prompting [22; 44; 45; 46], or training supervised learning [10; 31; 32; 47] or reinforcement learning (RL) [48; 49; 50; 16]. When LLMs are trained with the RL algorithm, the generation of LLMs can be naturally expressed as the Markov Decision Process (MDP) and optimized for specific objectives. According to this formula, InstructGPT [51] has achieved remarkable success in optimizing LLMs to align human preferences by utilizing RL from Human Feedback (RLHF) [52]. RLAIF then uses AI feedback to extend RL from human feedback [53]. Our work aims to propose an LLM self-training method via process rewards guided tree search.

## 5.2 Large Language Model Reasoning

LLM reasoning algorithms include prompt-based chain-of-thought (CoT) [22], planning-based represented by tree-of-thought (ToT) [24]. Scientific reasoning has several categories to mine the potential of existing large language models, resulting from different performances for problem-solving. Previous studies have attempted to outperform the direct generation. For example, in this paper [54], an approach for generating solutions in a step-by-step manner is proposed, another model or function is used to select the top-ranked answers, and hallucination is avoided by limiting the output to a narrower set. [55] presents a maieutic prompting inference method, which can generate abductive explanations of various hypotheses explained by recursion, eliminate contradicting candidates, and achieve logically consistent reasoning. Chain-of-thoughts (CoT) [22] imitates the thought process like humans to provide step-by-step solutions given a question. Self-Consistency CoT [23] improves the reliability and Self-Consistency of answers by sampling multiple interpretations from LM and then selecting the final answer that appears most frequently. Tree-of-Thoughts (ToT) [24] further generalizes the CoT methodology by considering multiple different reasoning paths in the tree and exploring coherent units of thought to execute thoughtful decision-making. In our work, we benchmark hard science reasoning tasks against [22; 24; 34; 35].

# 6 Conclusion

In this paper, we propose ReST-MCTS*, self-training both policy and process reward model by high-quality samples generated by reward guided tree search. Inferred rewards from the previous iteration are able to refine the process reward model and self-train the policy model with high-quality traces. Experimental results show that the ReST-MCTS* outperforms other self-training paradigms and achieves higher accuracy than previous reasoning baselines under the same search budget.

**Limitation:** We discussed limitation in detail at Section H in Appendix. In summary, we need to show the ReST-MCTS* can generalize to other reasoning tasks outside of math (like coding, agent, etc); and tasks without ground-truth (dialogue, SWE-Bench [56], etc). We also need to scale up the proposed value model and further improve the data filtering techniques. One potential idea is to incorporate online RL algorithms that can help perform better self-training for value models and policy models.

## Acknowledgments

Dan and Sining would like to thank Zhipu AI for sponsoring the computation resources used in this work. Yisong is supported in part by NSF #1918655. Yuxiao and Jie are supported in part by the NSFC 62276148, NSFC for Distinguished Young Scholar 62425601, a research fund from Zhipu, New Cornerstone Science Foundation through the XPLORER PRIZE and Tsinghua University (Department of Computer Science and Technology) - Siemens Ltd., China Joint Research Center for Industrial Intelligence and Internet of Things (JCIIOT). Corresponding authors: Yisong Yue, Yuxiao Dong, and Jie Tang.

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

# Appendix

## Table of Contents

## A  Preliminaries

In this section, we briefly describe LLM reasoning, reward verification, and LLM self-training. The definitions for notations are in Table 5 and model comparison in Figure 6.

### A.1  LLM Reasoning

The use of reasoning approaches can significantly improve LLM problem-solving abilities [10]. Given a policy model, $\pi$ (an autoregressive pre-trained language model) and an input problem $Q$, $\pi$ can autoregressive generate an output sequence $s = (s_1, s_2, \cdots, s_K)$ by predicting the next token. The conditional probability distribution of generating the complete output sequence is:

$$\pi(s|Q) = \prod_{k=1}^{K} \pi(s_t|s_{<t}, Q).  \tag{3}$$

Table 5: Notation Table.

| Character | Meaning |
|---|---|
| $Q$ | given question/problem |
| $A$ | decoded answer |
| $a^*$ | final correct answer |
| $s$ | solution |
| $p$ | partial solution |
| $s_k$ | $k$-th step of solution $s$ |
| $K$ | number of reasoning steps of a solution |
| $A_j$ | $j$-th solution |
| $N$ | number of solutions |
| $d$ | number of preference pairs |
| $r_{s_k}$ | common PRM reward of a single step $s_k$, used to define weighted reward |
| $w_{s_k}$ | weighted reward of a single step $s_k$, inferred in self-training process after trace generation |
| $v_k$ | quality value of partial steps $p_k$, used to guide search; inferred in self-training process |
| $m_k$ | reasoning distance of partial steps $p_k$ |
| $\pi_B$ | base language model |
| $D_{S_0}$ | original training dataset |
| $V_\theta$ | process reward model |
| $r_\phi$ | outcome reward model |

Any problem can be reasoned by zero-shot prompting, few-shot prompting [57], chain-of-thought (CoT) [22], self-consistency CoT [23] or best-of-N (BoN) selection [1], tree-of-thought (ToT) [24], Monte Carlo tree search (MCTS) [14], graph-of-thought (GoT) [58], amongst other approaches. Generally, recent studies represented by CoT [22] aim to improve the overall performance as follows:

$$P_\pi(A = a^* \mid Q) = \mathbb{E}_{(s_0, s_1, \cdots, s_K) \sim P_\pi(s|Q)} \Big[ P(A = a^* \mid s_0, s_1, \cdots, s_K, Q) \Big]. \tag{4}$$

We often call each trajectory $(s_1, s_2, \cdots, s_K)$ a reasoning trace. $P(A = a^* \mid s_0, s_1 \ldots, s_K, Q)$ is the probability to get correct answer $a^*$ given a problem $Q$ and a reasoning trace $s$. Given a original training dataset $D = \{Q_1, Q_2, \cdots, Q_M\}$, a new dataset can be produced by sampling $\pi$ $N$ times per problem Q using the above-mentioned reasoning strategies:

$$D_{S_0} = \{(Q_1^j, s_1^j)|_{j=1}^N, \cdots, (Q_M^j, s_M^j)|_{j=1}^N\}. \tag{5}$$

As shown in Table 1, STaR [4], RFT [5], ReST$^{EM}$ [6], V-STaR [7], and Self-Rewarding [16] adopt CoT prompting. Step-by-step [1] and MATH-SHEPHERD [12] leverage the best-of-N selection as a reasoning evaluation strategy. TS-LLM [14] utilizes MCTS as a reasoning policy to fully generate traces. Our work similarly seeks a correct reasoning path to maximize the expected cumulative $P$.

### A.2 Reward Verification

In the context of LLMs, we assume that the reasoning trajectory is derived from the policy model $\pi$ sampling. The common first step of self-training methods is fine-tuning the base model $\pi$ on the original dataset $D_{S_0}$ and obtaining a new generator $\pi_{S_0}$. Besides, a reward $r$ is considered to evaluate the value of the history trace. The objective of reinforcement learning (RL) with reward $r(Q, s)$ is:

$$\mathcal{L}_{\text{RL}} = \mathbb{E}_{Q \in D_{S_0}} [\mathbb{E}_{s \in \pi(s|Q)} [r(Q, s)]]. \tag{6}$$

Recent works [1; 12], through PRMs and ORMs, model the objective of reasoning as a search to find the highest cumulative reward trajectory $s$ to a problem $Q$ and infer the final answer $A$.

• **ORM** ($D \times S \to \mathbb{R}$) is trained with a cross-entropy loss:

$$\mathcal{L}_{\text{ORM}} = A_s \log r_s + (1 - A_s) \log(1 - r_s), \tag{7}$$

where $A_s$ is the golden answer ($A_s = 1$ if $s$ is correct else $A_s = 0$) and $r_s$ is the ORM's output sigmoid score. STaR [4], RFT [5], and ReST$^{EM}$ [6] consider the outcome reward as value label $A_s$ compared with ground truth answer. In **V-STaR** [7], its outcome reward is generated by multi-iteration LLMs, and reward $r$ is defined via verifier $\pi_V$ and LLM generator $\pi_{S_0}$ as follows:

$$r(Q, s) = \beta \log \frac{\pi_V(s|Q)}{\pi_{S_0}(s|Q)}. \tag{8}$$

$\beta$ is the hyper-parameter that controls the proximity of the reference policy $\pi_{S_0}$. Different from V-STaR, the outcome rewards $r$ in self-rewarding [16] is generated through LLM-as-a-Judge prompting using dataset $D_{S_0}$ and policy model $\pi_{S_0}$. In both V-STaR and self-rewarding, collected $D_{\text{VER}}$ is built for training verifiers as follows:

$$D_{\text{VER}} = \{(Q_j, s_{j,1}^+, s_{j,1}^-), \cdots, (Q_j, s_{j,d}^+, s_{j,d}^-)\}_{j=1}^N, \tag{9}$$

$d$ is the number of preference pairs. However, previous works [1; 12] suggested PRMs demonstrate better supervision than ORMs among false positive solutions and provide more reliable feedback.

- **PRM** ($D \times S \to \mathbb{R}^+$) is trained with:

$$\mathcal{L}_{\text{PRM}} = \sum_{i=1}^K A_{s_k} \log r_{s_k} + (1 - A_{s_k}) \log(1 - r_{s_k}), \tag{10}$$

where $A_{s_k}$ is the golden answer ($A_{s_k} = 1$ if $s_k$ is correct else $A_{s_k} = 0$) of $s_k$ and $r_{s_k}$ is the PRM's output sigmoid score. Specifically, [1] regards PRM training as a three-class classification with **costly human annotations**. Similarly, MATH-SHEPHERD [12] collects random rollout trajectories via BoN reasoning policy and synthesizes process rewards to construct the PRM training dataset autonomously. MATH-SHEPHERD defines the **automated quality** $r_{s_k}$, the potential to deduce the correct answer, for each reasoning step $s_k$ through hard estimation (HE) and soft estimation (SE), which are,

$$A_{s_k}^{\text{HE}} = \begin{cases} 1, & \exists A_j \in A^*, A_j = a^* \\ 0, & \text{Otherwise} \end{cases}, \tag{11}$$

$$\text{and} \quad A_{s_k}^{\text{SE}} = \frac{\Sigma_{j=1}^N \mathbb{I}(A_j = a^*)}{N}. \tag{12}$$

$A^* = \{A_j\}_{j=1}^N$ and $N$ is the number of solutions. To more precisely find a reasoning trajectory with the highest expected reward, RAP [59] utilizes MCTS to estimate the **expected future reward** via state-action function ($\mathcal{Q} : \mathcal{S} \times \mathcal{A} \longmapsto \mathbb{R}$) in traditional RL for each node. However, the decoding process of incorporating MCTS into LLMs is costly, because estimating and updating a state-action function requires a recursive visit, and reward values need to be calculated by LLMs. The concurrent work pDPO [13], though effective, does not take internal accuracy generated by LLMs into account and ignores the number of steps to be generated. In this paper, we integrate process reward guidance with tree search to explore efficient solution space and synthesize high-quality trajectories.

### A.3 LLM Self-Training

**Generation.** Given a new training dataset $D_G$, self-training methods use generator $\pi_{S_0}$ to generate reasoning steps $s$ and final answer $A$ per problem $Q$. In each iteration $i$ ($i \geq 1$), STaR, RFT, and ReST$^{\text{EM}}$ check the generated solutions $D_{G_i}$ with the binary correctness label $z$ and keeps the correct solutions $(A_j = a^*)|_{j=1}^N$ as $D_{G_i(A_j=a^*)|_{j=1}^N}$. Based on the continuous iteration on positive samples, V-STaR and Self-Rewarding keep the correct and incorrect generated solutions per problem $Q$ and train preference data pairs on constructed verifier data $D_{\text{VER}}$ with all data $D_{G_i}$, so the $\pi_V$ can learn the error patterns produced by a generator in each iteration $i$. Then, the generator $\pi_{S_{i-1}}$, here is $\pi_{S_0}$, is fine-tuned on new generated dataset $D_{G_i(A_j=a^*)|_{j=1}^N}$ and again is updated as generator $\pi_{S_i}$. This process is continuously running in subsequent iterations. Their iterative process and reward value are as follows:

$$D_{S_0} \xrightarrow{\pi} \pi_{S_0} \xrightarrow{D_G} D_{G_i(A_j=a^*)} \xrightarrow{\pi_{S_{i-1}}} \pi_{S_i}, \tag{13}$$

where $i = 1$ for RFT and $i \geq 1$ for STaR, ReST$^{\text{EM}}$, V-STaR, and Self-Rewarding.

**Improvement.** The practical way to accomplish reasoning tasks on $D_{S_0}$ is supervised fine-tuning (SFT) that trains a policy model by minimizing the negative log-likelihood loss on the training dataset:

$$\mathcal{L}_{\text{SFT}}(\pi) = -\mathbb{E}_{(Q,s)\in D_{S_0}} \Big[ \sum_{t=1}^T \log \pi(s_t | s_{<t}, Q) \Big]. \tag{14}$$

Recent offline preference learning methods replace LLM verifiers (before being trained on LLM generator and binary classification) with DPO [50]. The training DPO objective for a verifier $\pi_V$ is described as follows:

$$\mathcal{L}_{\text{DPO}}(\pi_V; \pi_{S_0}) = -\mathbb{E}_{(Q,s^+,s^-)\sim D_{\text{VER}}} [\log \sigma(r(Q, s^+) - r(Q, s^-))], \tag{15}$$

where $\sigma$ is the logistic function.

# B Deduction Demonstration

## B.1 Definition of Weighted Value and Quality Value

**Weighted Value.** Recall the definition of the weighted reward:

$$w_{s_k} = \frac{1 - v_{k-1}}{m_k + 1}(1 - 2r_{s_k}), \ \ k = 1, 2, \cdots \tag{16}$$

And we know that $r_{s_k} \in [0, 1]$. Now, let's examine the maximum possible value of the term $(1 - 2r_{s_k})$. Since $r_{s_k} \in [0, 1]$, the maximum value of $(1 - 2r_{s_k})$ occurs when $r_{s_k} = 0$. In this case, $(1 - 2r_{s_k}) = 1$. Therefore, we can conclude that $-1 \le (1 - 2r_{s_k}) \le 1$.

Next, let's consider the denominator, $(m_k + 1)$. Since $m_k = K - k$, and $K \ge k$, we have $m_k \ge 0$ and $m_k + 1 \ge 1$. Therefore, we can conclude that $(m_k + 1) \ge 1$.

Combining these results, we can rewrite the weighted reward as follows:

$$w_{s_k} = \frac{1 - v_{k-1}}{m_k + 1}(1 - 2r_{s_k}) \le |1 - v_{k-1}| \cdot |1 - 2r_{s_k}| \le |1 - v_{k-1}| \tag{17}$$

Hence, we deduce that $w_{s_k} \le |1 - v_{k-1}|$, which indicates that the weighted reward is bounded by the absolute value of the difference between 1 and the previous quality value.

**Quality Value.** Recall that the quality value $v_k$ is determined by incorporating the previous quality value $v_{k-1}$ and the weighted reward $w_{s_k}$ of the current step.

$$v_k = max(v_{k-1} + w_{s_k}, 0) \tag{18}$$

Now, we can inductively reach the conclusion that $v_k \in [0, 1]$, starting from the fact that $v_0 = 0$. Assuming that $v_{k-1} \in [0, 1]$, then we can derive $v_k \in [0, 1]$ using the bound of $w_{s_k}$.

$$v_k = max(v_{k-1} + w_{s_k}, 0) \le v_{k-1} + |1 - v_{k-1}| = 1 \tag{19}$$

Therefore, based on the properties of the weighted reward and the definition of the quality value, we can deduce that $v_k$ is indeed confined within the range $[0, 1], k = 0, 1, 2, \cdots$.

**Fine-grained Dataset for Math.** We adopt an alternative method to generate value train data for math. For this method, we only demand a correct final answer $a_*$ for each question $q$, which is simpler to satisfy. Specifically, we integrate the MATH [33] train set into $D_0$. For each question $q^{(i)}$ and answer $a_*^{(i)}$, we use Mistral-7B: MetaMATH [31] as a policy to generate solution traces in a simple breadth-first-search (BFS) manner, obtaining a search tree $T_q^{(i)}$ similar to the one of the self-training process. Subsequently, we verify the obtained answers of all leaf nodes of $T_q^{(i)}$ according to $a_*^{(i)}$. The verified search trees are then used to derive data samples with target values for $D_{V_0}$.

• **Construction of value model training set.** Previous approaches like [1] that employ PRMs usually require human annotation to initialize a train set, which is quite costly. In comparison, our value model's initial training set can be constructed at a lower expense.

For math data, we deploy the same approach mentioned in section 3.2 to infer process rewards and quality values of partial solutions within the verified search tree $T_q^{(i)}$. While for science data, this value-inferring process is slightly different. We still derive the target value of $p_k^{(i)}$ based on the definition in Eq. (1) and Eq. (2). Under the assumption that original solutions are reliable and concise, we can simply regard $s^{(i)}$ as the globally optimal reasoning path for $q^{(i)}$. Therefore, we derive that:

$$r_{s_k}^{(i)} = 0, m_k^{(i)} = K_i - k, w_k^{(i)} = \frac{1}{K_i} \text{ and } v_k^{(i)} = \frac{k}{K_i}. \tag{20}$$

*Derivation.* Please refer to the detailed derivation for Eq. (20) in Appendix B.2.

In contrast, for generated false samples, we set $r_{s_{k+1}}^{(i)'} = 1, m_{k+1}^{(i)'} = K_i - k$ (since still $K_i - k$ correct reasoning steps required to reach final answer). Considering that $v_k^{(i)} = \frac{k}{K_i}$, we have:

$$w_{k+1}^{(i)'} = -\frac{K_i - k}{K_i - k + 1}\frac{1}{K_i}, \tag{21}$$

$$v_{k+1}^{(i)'} = \max(0, \frac{k-1}{K_i} + \frac{1}{K_i \cdot (K_i - k + 1)}). \tag{22}$$

*Derivation.*   Please refer to the detailed derivation for Eq. (21) and Eq. (22) in Appendix B.2.

Collecting all samples and their corresponding derived quality values, we acquire the initial training set $D_{V_0}$ for value model $V_\theta$, as described in Appendix E.1.

## B.2   Detailed Deduction for Weighted Value and Quality Value

Here, we deduce the weighted reward using Eq. (2) and quality value using Eq. (1):

when $k = 0$:

$$v_0 = 0 \tag{23}$$

when $k = 1$:

$$w_1 = \frac{1-0}{K-1+1}(1 - 2 \times 0) = \frac{1}{K} \tag{24}$$

$$v_1 = \max(0 + \frac{1}{K}, 0) = \frac{1}{K} \tag{25}$$

when $k = 2$:

$$w_2 = \frac{1 - \frac{1}{K}}{K - 2 + 1}(1 - 2 \times 0) = \frac{K-1}{K(K-1)} = \frac{1}{K} \tag{26}$$

$$v_2 = \max(\frac{1}{K} + \frac{1}{K}, 0) = \frac{2}{K} \tag{27}$$

therefore,

$$w_k = \frac{1}{K}, w_{k-1} = \frac{1}{K}, w_{k+1} = \frac{1}{K} \tag{28}$$

$$v_k = \frac{k}{K}, v_{k-1} = \frac{k-1}{K}, v_{k+1} = \frac{k+1}{K} \tag{29}$$

$$m_k = K - k, m_{k-1} = K - k + 1, m_{k+1} = K - k - 1. \tag{30}$$

Then, we deduce the Eq. (21) and Eq. (22):

$$
\begin{aligned}
w_{k+1} &= \frac{1 - v_k}{m_{k+1} + 1}(1 - 2r_{s_{k+1}}) \\
&= \frac{1 - \frac{k}{K}}{(K-k) + 1}(1 - 2 \times 1) \\
&= -\frac{K-k}{K(K-k+1)} \\
&= -\frac{1}{K}\frac{K-k}{K-k+1}
\end{aligned} \tag{31}
$$

$$
\begin{aligned}
v_{k+1} &= \max(v_k + w_{k+1}, 0) \\
&= \max(\frac{k}{K} - \frac{1}{K}\frac{K-k}{K-k+1}, 0) \\
&= \max(\frac{k(K-k+1) - (K-k)}{K(K-k+1)}, 0) \\
&= \max(\frac{k(K-k+1) - (K-k+1) + 1}{K(K-k+1)}, 0) \\
&= \max(\frac{(K-k+1)(k-1) + 1}{K(K-k+1)}, 0) \\
&= \max(\frac{(K-k+1)(k-1)}{K(K-k+1)} + \frac{1}{K(K-k+1)}, 0) \\
&= \max(\frac{k-1}{K} + \frac{1}{K(K-k+1)}, 0)
\end{aligned}
\tag{32}
$$

## C    Algorithm Detail and Process Example

### C.1    Algorithm Details of MCTS*

**Node Selection.** Similar to [14], we propose to start each selection process from the initial root, since this allows backtracking. Within each iteration, the node selection stage is first executed, where a leaf node $C_{select}$ is hierarchically selected starting from the initial root. To incorporate the quality value of nodes, we use UCB as the criterion to select a child rather than the UCT [26], which is as follows:

$$
UCB(C) = v_C + \epsilon\sqrt{\frac{\ln n_{parent}}{n_C}},
\tag{33}
$$

where $n_{parent}$ is the number of visits of the parent node of $C$, $\epsilon$ is a exploration constant. For each intermediate node, we select its child with maximum UCB. This criterion considers both quality value and visit count, thus it encourages the exploration of high-quality nodes while leaving some opportunity for underexplored nodes.

**Thought Expansion.** Secondly, the value of the selected node $C_{select}$ is compared with a threshold $l$ (in our experiments, $l$ it is set to 0.9). If the $v_{C_{select}} >= l$, the node's recorded partial solution $p_{C_{select}} = [s_1, s_2, \cdots, s_k]$ is deemed acceptable as a final solution (since $v_C$ get close to 1 only when $C$ is close to correct final answer), which is then directly returned as output, terminating the algorithm. This is different from the method adopted by [60] since no reward model estimation is required. Otherwise, the expansion stage is initiated, where new solution steps $s_{k+1,i}(i = 1, 2, \cdots, b)$ are sampled by prompting the policy $\pi_{S_0}$, i.e. $s_{k+1,i} \sim \pi_{S_0}(s_{1,2,\cdots,k}|q)$, $b$ is the number of samples or branches. Subsequently, new nodes $C_i = ([s_1, s_2, \cdots, s_k, s_{k+1,i}], 0, v_{C_i})$ are added to $T_q$ with $v_{C_i}$ assigned by the value model, $v_{C_i} \leftarrow V_\theta(p_{C_i}|q)$. Note that we also incorporate a self-critic mechanism into this expansion process, which will be illustrated later.

**Greedy MC Rollout.** [60] and [14] use a simplified three-stage iteration that doesn't include a simulation process on leaf nodes. In contrast, we believe that a simulation process still brings about useful information for value estimation, despite the rise in generation and time cost. In this stage, we propose to simulate a few steps upon the new node $C_i$ with maximum predicted value. Reasoning steps starting from this node will be sampled step-by-step and evaluated, while only the most valuable path is further explored, until a step limit $m$ is reached. The highest quality value acquired in the sampling process $v_{max}$ is recorded and used to update $v_{C_i}$ with a weight parameter $\alpha$ following:

$$
v_{C_i} \leftarrow \alpha v_{C_i} + (1 - \alpha)v_{max}.
\tag{34}
$$

Besides, the visit count $n_{C_i}$ is also updated by $n_{C_i} \leftarrow n_{C_i} + 1$.

**Value Backpropagation.** Finally, we conduct value backup starting from $C_{select}$. The value of every parent node of $C_{select}$ is updated using a weighted average method. For every node $C$ on the trace from root to $C_{select}$, we update its $n_C$ and $v_C$ as follows:

$$
n_C \leftarrow n_C + 1
\tag{35}
$$

**Algorithm 2:** The proposed value guided search algorithm MCTS*.

---

**Input:** question $q$, inference_model $\pi_{S_0}$, value_model $V_\theta$, max_iterations $T$, threshold $l$, branch $b$, rollout_steps $m$, roll_branch $d$, weight_parameter $\alpha$.

1: $T_q \leftarrow$ Initialize_tree$(q)$
2: $\pi_{S_0}, V_\theta \leftarrow$ Initialize_models$(\pi_{S_0}, V_\theta)$
3: **for** $i$ in range$(T)$ **do**
4:    $C \leftarrow$ root$(T_q)$
5:    ————————*Node Selection*————————
6:    **while** $C$ is not leaf node **do**
7:       $C \leftarrow argmax_{C' \in \text{children}(C)}(v_{C'} + \epsilon\sqrt{\frac{\ln n_C}{n_{C'}}})$ // Select child node based on UCB
8:    **end while**
9:    **if** $v_C \geq l$ **then**
10:      **Return** $p_C$ // Output solution
11:    **end if**
12:    ————————*Self-Critic*————————
13:    $o \leftarrow$ Do_self_critic$(p_C|q, \pi_{S_0})$ // Get $\pi_{S_0}$ response for further exploitation or stop reasoning
14:    **if** $o$ is not **EoI then**
15:      ————————*Thought Expansion*————————
16:      **for** $j$ in range$(b)$ **do**
17:        $C_j \leftarrow$ Get_new_child$(o, p_C|q, \pi_{S_0})$ // Expand based on previous steps and self-critic
18:        $v_{C_j} \leftarrow V_\theta(p_{C_j}|q)$ // Evaluate with value model
19:      **end for**
20:      ————————*Greedy MC Rollout*————————
21:      $C' \leftarrow argmax_{C' \in \text{children}(C)}(v_{C'})$
22:      $p = p_{C'}$
23:      $v_{max} = 0$
24:      **for** $k$ in range$(m)$ **do**
25:        $p, v_{max} \leftarrow$ Get_next_step_with_best_value$(p|\pi_{S_0}, V_\theta, d, q)$ // Sample new children and record the best-observed value
26:      **end for**
27:      $v_{C'} \leftarrow \alpha v_{C'} + (1 - \alpha)v_{max}$
28:      $n_{C'} \leftarrow n_{C'} + 1$ // Update value and visit count of the rollout node
29:    **end if**
30:    ————————*Value Backpropagation*————————
31:    Back_propagate$(C)$ // Update value of parent nodes using weighted average
32: **end for**
33: $C=$ Get_best_node$(T_q)$ // Fetch the node with the highest value in the search tree
34: **Return** $p_C$

**Output:** $p_C$

---

and

$$v_C \leftarrow \frac{\Sigma_i n_{C_i} \cdot v_{C_i}}{\Sigma_i n_{C_i}} \tag{36}$$

where $C_i(i = 1, 2, \cdots, b)$ are the children of $C$. This actually updates the value of $C$ according to its children's value expectation.

**Determine Termination via Self-Critic.** Although the value model provides accurate evaluation for partial solutions, it cannot consistently signal logical termination, especially when the inference model reaches a false conclusion. Consequently, even when a false final answer is generated, further exploration beneath this node may still be conducted, leading to reduced search efficiency. Therefore, we propose to use self-critic to provide extra timely signals of logical termination that avoid unwise exploration as well as insight into deeper search. Specifically, we prompt the inference model to generate an **End of Inference** (EoI) signal or offer advice $o$ on following exploration steps based on existing partial solutions $p$ before each expansion stage. The expansion and MC rollout stage will be

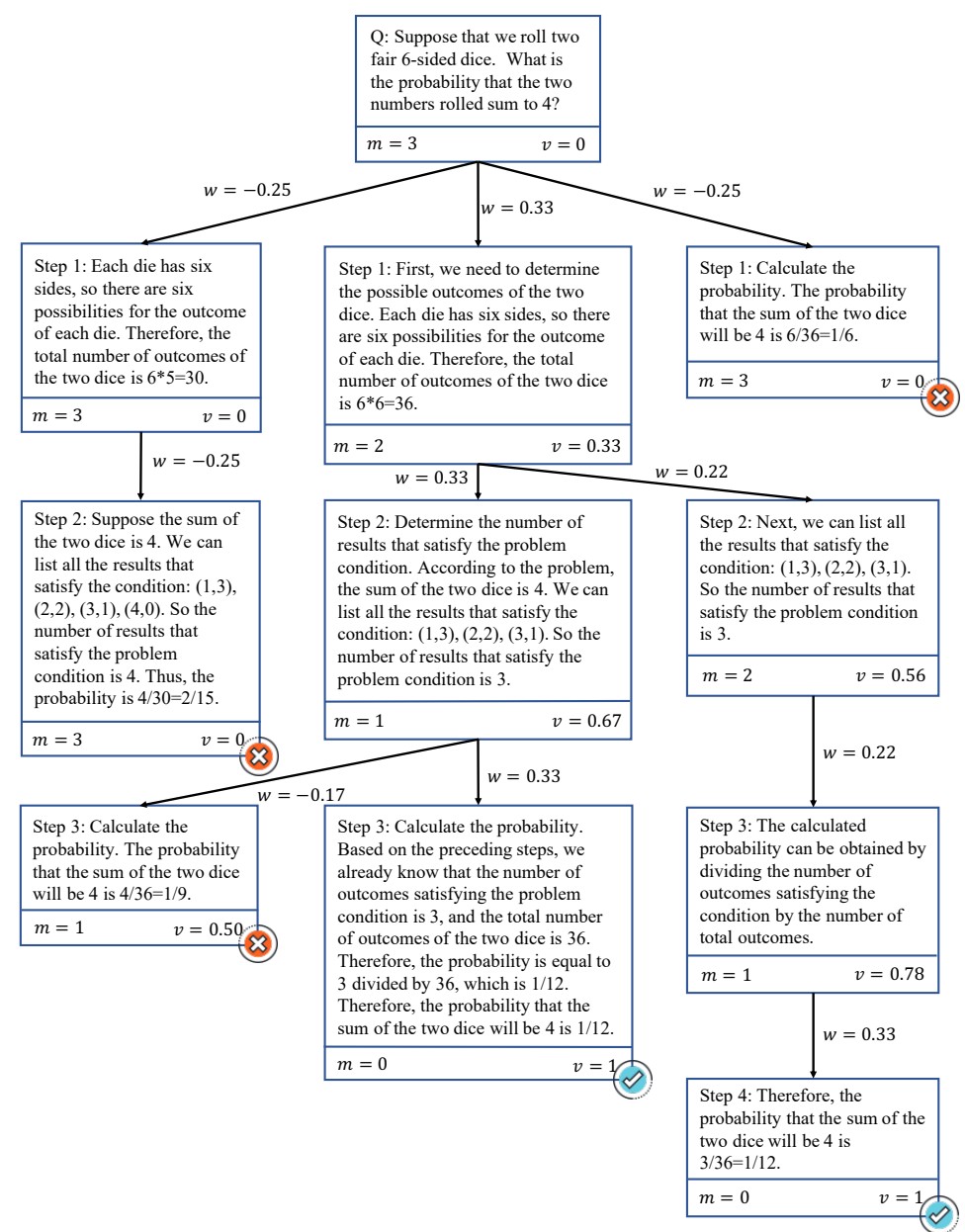

Figure 3: Detailed inferred process of a concrete example. The search tree $T_q$ has already been pruned during this stage, with traces verified. Starting with the inference of $m$, we gradually update all weighted rewards $w$ for actions (steps) and quality value $v$ for states (partial solutions). Taking the false Step 3 as an example, since it makes a mistake in calculation, it still requires the same number of steps as its parent to reach the correct answer (one step to correct the calculation mistake), i.e. $m = 1$. As no trace starting from this node reaches the correct answer, we have $r_s = 1$. Thus, we derive $w = \frac{1-0.67}{1+1}(1 - 2 \cdot 1) = -0.17$, and therefore $v = \max(0.67 + (-0.17), 0) = 0.50$.

skipped if an **EoI** signal is received. Otherwise, the advice will be utilized in the following expansion stage as part of the inference prompt, so $\pi_{S_0}$ generates new steps based on both $o$ and $p$. An overall pseudo-code for ReST-MCTS$^*$ is presented in Algorithm 2.

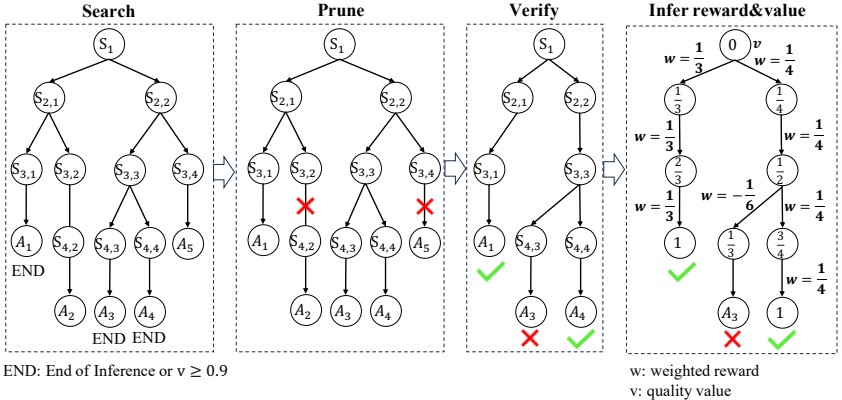

Figure 4: Detailed process of new sample data generation for the self-training framework.

## C.2 Data Generation Process and Specific Example for Reward Inference

The data generation process of our self-training approach consists of mainly four stages, namely search, prune, verify, and reward inference, which is demonstrated in Figure 4. For reward inference, a detailed example is shown in Figure 3.

## D Model Comparison

**ReST-MCTS\* vs. AlphaLLM.** As an approach that aims to enhance LLM inference, AlphaLLM [61] utilizes a tailored MCTS algorithm and critic models to provide precise feedback. Even though AlphaLLM also adopts MCTS and critic models for self-improvement, their approach is different from ours in various crucial aspects, as elaborated below.

**(1) Design of MCTS algorithm.** For the level of search, AlphaLLM's $\eta$MCTS considers options as action, with termination signals delivered by a termination function $\beta$. In contrast, we use reasoning steps as action, which is achieved through tailored prompt design. Concerning critic models, we use a single value model to provide evaluation for intermediate nodes. The model is trained to predict specially designed quality values that reflect completeness and correctness of partial solutions, rather than estimating the conventional definition of value function in RL. In addition, we also incorporate self-critic mechanisms into the tree search algorithm to provide insights for the policy (Appendix C.1), which AlphaLLM does not adopt.

**(2) Definition of reward/value.** Our definition of weighted reward and quality value is novel, leading to significant differences between our method and AlphaLLM across various processes such as critic model training, data synthesizing, and data filtration. Since our design of quality value involves information on process reward and reasoning distance, our value model trained on this target can naturally provide sufficient feedback during the search, with no need for implementing other critic models mentioned by AlphaLLM.

**(3) Self-Training algorithm.** Although AlphaLLM also includes iterative self-training, the implementation method varies greatly. Most importantly, their critical model is static throughout the iterations, which means they focus more on the improvement of policy. In comparison, we also consider the impacts of self-training on the critic value model. As demonstrated in Algorithm 1, we calculate process rewards and quality values according to the final search tree of questions within each iteration, which are then used as new training data for the value model.

## E Experimental Details

### E.1 Training and Evaluation of Initial Value Model

**Initialization of Value Model.** We split $D_{V_0}$ and use the train set to finetune ChatGLM3-6B and Mistral-7B to predict the value of partial solutions. We simply add a linear layer to the model to

directly transform probabilities to a scalar value. Moreover, we use the AdamW optimizer [62] and MSE loss in Eq. (37) to optimize, eventually obtaining an initial value model $V_\theta$ that can evaluate the correctness and completeness of step-by-step solutions. Note that the learning rate is set to 1e-6 in this process. The MSE training loss is shown below:

$$L_{\text{MSE}} = E_{(q,p,v) \sim D_{V_0}} |V_\theta(p|q) - v|^2. \tag{37}$$

**Evaluation of Value Model.** We use the test set containing 14k data samples to evaluate the value model with an absolute tolerance of 0.1:

$$\text{Accuracy} = \frac{1}{t} \Sigma_{i=1}^{t} \ I(|clip(V_\theta(p_i|q_i), 0, 1) - v_i^*| < 0.1) \tag{38}$$

where $t$ is the number of test data samples, $q_i$ is the question of sample $i$, $p_i$ is the partial solution of sample $i$ and $v_i^*$ is the target value of sample $i$. Our initial value model achieves an accuracy of **69.3%**, which means it is reliable in most situations. We also conducted a study to measure the value model's performance on science benchmark SciBench [34] compared to outcome-supervised reward models and self-critic methods in Table 7.

### E.2 Benchmark Setup

To compare the performance of different search methods, we construct a standardized benchmark test that can be generally used on labeled science or math datasets like MATH, SciBench, and SciEval. Aside from the ReST-MCTS*, we incorporate two other baselines: chain-of-thought (CoT) and tree-of-thought (ToT). For each method, specialized prompts $P$ are designed to execute the search process. Besides, an inference model $\pi$ and value model $V$ are deployed to provide deduction and feedback. Concerning the CoT baseline, we use Self-Consistency to calculate accuracy. For the ToT baseline, we use a simple greedy depth-first-search (DFS) algorithm with node values assigned by the value model. The algorithm stops exploitation when a max depth of 10 is reached and ends when a node value exceeds the threshold 0.9. For ReST-MCTS*, self-critic is used and the ending threshold is also set to 0.9. The rollout step limit $m$ is set to 2, $\alpha$ is set to 0.5, and the number of iterations $T$ is set to 50 by default. Moreover, both tree search algorithms use $b = 3$ by default, where $b$ is the number of samples generated in the expansion process as mentioned in the former sections. After the search process, the policy is prompted to extract the final answer based on the obtained solution, which is then compared with the ground truth to determine correctness. The results of these methods on benchmarks are illustrated in Section 4.3.

### E.3 Baselines of Search Verification

The basic settings of relevant verification baselines are illustrated as follows:

- **ORM + Best-of-N (BoN)** For simplicity, we employ the ORM used by [10], which is trained on SciInstruct. For each question, we sample $N$ solutions and select the solution with the highest ORM score as output. $N$ is used to control token usage.

- **ReST-MCTS*** Implementation of ReST-MCTS*, using the value model $V_\theta$ as PRM to guide MCTS*. The variable controlling token usage is the iteration number $T$ and branch parameter $b$.

- **Self-Consistency** $N$ solutions are generated for each question using a simple CoT prompt. Their final answers are then extracted and classified, with the most frequently occurring answer selected as the final output. $N$ is used to control token usage.

- **PRM + Best-of-N (BoN)** With value model $V_\theta$ used as PRM, we perform DFS-based tree search. Every selected solution $s$ is evaluated by a PRM score $r_{\text{PRM}} = \Pi_{i=1}^{K} v_i$. The one with the highest PRM score among all $N$ solutions is regarded as the final output. Under this setting, $b$ is set to 3, while $N$ is used to control token usage.

As shown in Figure 2, we have compared the number of tokens consumed by each algorithm to achieve certain accuracy on MATH and SciBench. Results reveal that to reach a certain expected accuracy, MCTS* basically requires fewer tokens than other algorithms like SC and ORM + BoN. This means that based on the same expected standards, MCTS* outperforms other algorithms while maintaining a reasonable computation cost. As for the actual running time, we have recorded

Table 6: The average running time of different algorithms (under our basic experiment settings) on a single question.

| Method | CoT + SC | ORM + BoN | PRM + BoN | MCTS* |
|---|---|---|---|---|
| Running time (s) | 41 | 43 | 73 | 108 |
| MATH | 37.0 | 33.5 | 34.5 | 41.5 |

Table 7: Accuracy of different reward/value models on the questions selected from SciBench, they all use MCTS* as search policy.

| Dataset | Models | ORM | PRM | Self-Rewarding | ReST-MCTS* (Value) |
|---|---|---|---|---|---|
| SciBench | GLM4 | 20.2 | 22.0 | 20.2 | **22.9** |
| | GPT-3.5-turbo | 12.8 | 17.4 | 13.7 | **20.2** |

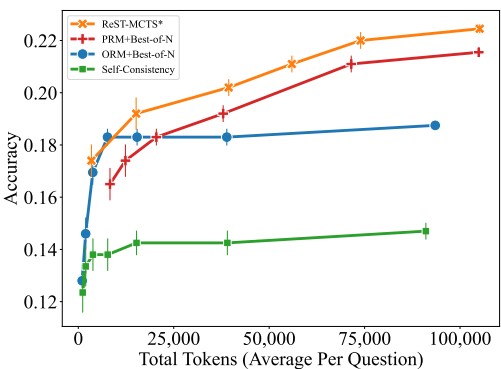

Figure 5: Accuracy of different methods on SciBench with varied total token usage per question. Both the completion token and prompt token are included.

the average running time of different algorithms (under our basic experiment settings) on a single question, as shown in Table 6. We see that MCTS* spends more time on exploration and simulation than other simple algorithms. However, since our method adopts a different design of value, it does not require massive Monte Carlo estimations. This reduces the running time of our algorithm and limits the time consumption to a reasonable range. Notice that MCTS* can achieve high accuracy that other algorithms can never attain even at unlimited cost, we believe this extra time is fairly acceptable.

### E.4 Value Model of ReST-MCTS* on SciBench

We employ the reward model obtained by [10] (which is used as a classifier for SciGLM) as the ORM and our fine-grained value model as PRM to provide the outcome reward and step-wise value respectively. We also include the Self-Rewarding method, where the policy model itself is instructed to provide step-wise value. For all methods, the number of samples for each step is set to 3. Using this setting, we record the model accuracy of GLM4 and GPT-3.5-turbo on the selected questions, which are as shown in Table 7. Results indicate that compared to ORM and Self-Rewarding, PRM-based methods exhibit higher accuracy. This confirms the effectiveness of our value model. In addition, Figure 5 concerns the total consumption of the token budget, including all prompt tokens and completion tokens. However, we still have to note that the total token usage (especially prompt tokens) of the ReST-MCTS* increases rapidly as hyper-parameters $b$ and $T$ rise.

### E.5 ReST-MCTS* on SciEval

**SciEval.** Similar to SciBench, we perform benchmark tests on SciEval. Results are shown in Table 8. For both GLM4 and GPT-3.5-turbo, ReST-MCTS* again outperforms other baselines in overall accuracy, with an accuracy of $79.87\%$ and $62.31\%$ respectively. However, we notice that though tree-search-based methods demonstrate an advantage on average, they fail to improve the performance of the CoT baseline on some parts of SciEval. We examine the data distribution and discover that

Table 8: Overall performance comparison with representative models on SciEval.

| Models | Method | Part I | Part II | Part III | Part IV | Ave. |
|---|---|---|---|---|---|---|
| GLM4 | CoT | 52.50 | 85.83 | 80.83 | 78.11 | 74.32 |
| | ToT | 60.83 | 87.77 | **84.16** | **79.77** | 78.13 |
| | ReST-MCTS* | **69.16** | **88.05** | 83.33 | 78.94 | **79.87** |
| GPT-3.5-Turbo | CoT | 28.88 | 78.61 | **70.00** | **71.46** | 62.24 |
| | ToT | **32.50** | 76.11 | 68.05 | 67.59 | 61.06 |
| | ReST-MCTS* | 29.72 | **78.88** | 69.72 | 70.91 | **62.31** |

these parts are basically all single-choice questions. As they are less difficult compared to other types of questions, the Self-Consistency CoT approach may already be competent. Besides, these questions often require few reasoning steps, which may be the main reason why tree search methods do not perform as well as expected.

# F    Prompt and Instruction Examples

We present some instruction examples used in ReST-MCTS* and self-training process in this section, including:

- **Inference instruction** This instruction is used in tree search for the policy to generate new steps based on previous self-critic information.

- **Self-critic instruction** Used for generating the **EoI** signal or advice for further search.

- **LLM verify instruction** This instruction is employed in the data generation process of self-training when an answer needs verification by LLM (GPT-4 for our case).

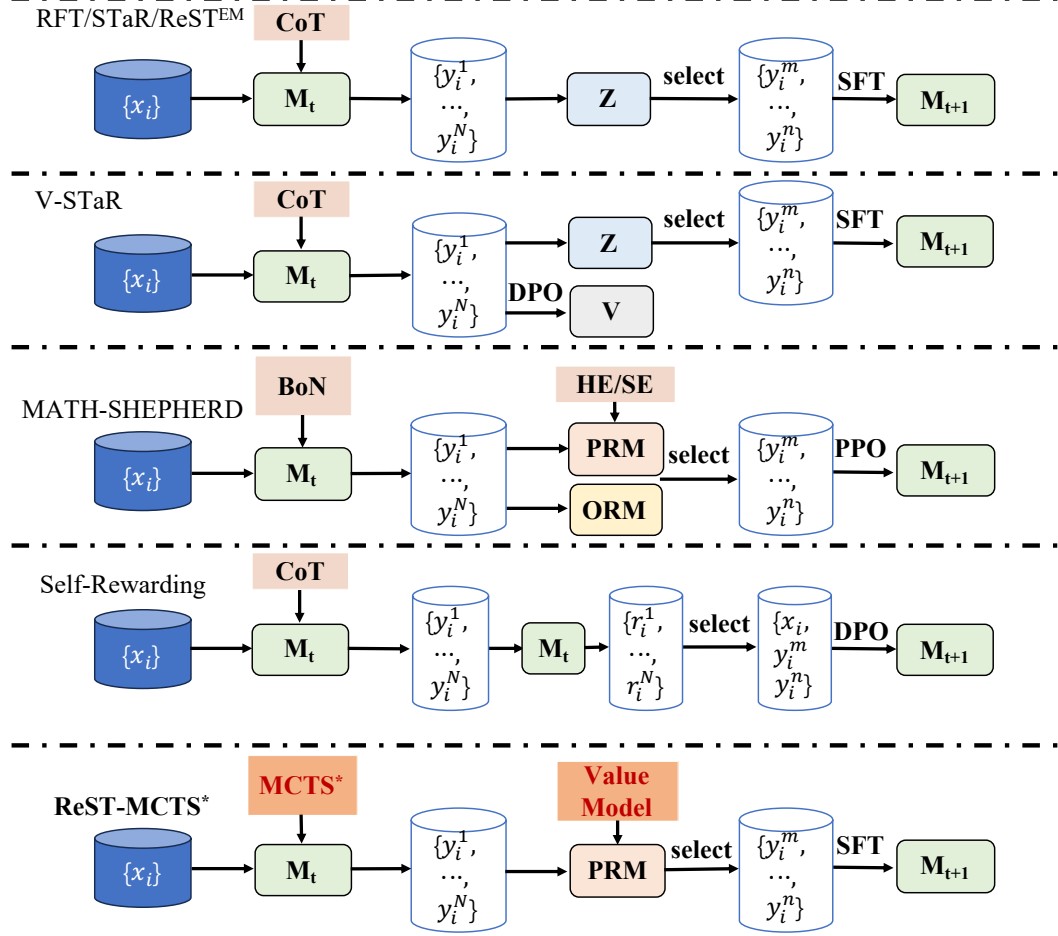

Figure 6: Comparison between existing self-training methods with our proposed ReST-MCTS$^*$.

---

**This is an example of inference instruction.**

**Instruction:** Given a science or math problem, a corresponding solution that may be incomplete as well as some advice on solving this problem, your task is to generate a new solution step based on the advice and existing steps.

**Problem:** Let $a$ and $b$ satisfy $ab = 7$ and $a + b = 5$. What is the value of $a^2 + b^2$?.

**Solution:**

Step 1: From $a + b = 5$, we can know that $(a + b)^2 = 25$.

Step 2: Thus, $a^2 + 2ab + b^2 = 25$.

**Advice:** From step 2 we know the value of $a^2 + 2ab + b^2$. Since we also know the value of $ab$, we can derive $a^2 + b^2$.

**Policy Output:** Since $a^2 + 2ab + b^2 = 25$ and $ab = 7$, we derive $a^2 + b^2 = (a^2 + 2ab + b^2) - 2ab = 25 - 14 = 11$.

---

**This is an example of self-critic instruction.**

**Instruction:** Given a science or math problem and a corresponding solution that may be incomplete, your task is to judge whether the solution has already reached a final answer or conclusion for the problem. If the solution has already reached a final answer or conclusion, you should directly output "Final answer reached". Otherwise, you should give some advice on how to solve the problem based on current steps or what to consider next.

**Problem:** Let $a$ and $b$ satisfy $ab = 7$ and $a + b = 5$. What is the value of $a^2 + b^2$?.
**Solution:**
Step 1: From $a + b = 5$, we can know that $(a + b)^2 = 25$.
Step 2: Thus, $a^2 + 2ab + b^2 = 25$.

**Policy Output:** The solution has not reached a final answer. Advice: From step 2 we know the value of $a^2 + 2ab + b^2$. Since we also know the value of $ab$, we can derive $a^2 + b^2$.

---

**This is an example of LLM verify instruction.**
**Instruction:** Given a science or math problem, a corresponding step-by-step solution, and the true answer of the problem, your task is to verify the answer obtained in the solution with the real answer. If the answer obtained in the solution is equivalent to the real one, output '1', otherwise output '0'.
**Problem:** Let $a$ and $b$ satisfy $ab = 7$ and $a + b = 5$. What is the value of $a^2 + b^2$?.
**Solution:**
Step 1: From $a + b = 5$, we can know that $(a + b)^2 = 25$.
Step 2: Thus, $a^2 + 2ab + b^2 = 25$.
Step 3: Since $ab = 7$, we can derive that $a^2 + b^2 = (a^2 + 2ab + b^2) - 2ab = 25 - 14 = 11$.
So the answer is 11.
**Real Answer:** The value of $a^2 + b^2$ is 11.

**LLM Output:** 1

# G  Further Preliminaries of MCTS and LLM Reasoning with MCTS

## G.1  Monte Carlo Tree Search (MCTS)

**MCTS** [63] is a search algorithm for optimal decision-making in large and complex combinatorial spaces. This algorithm represents search spaces as search trees and works on the principle of the best-first search based on the evaluations of stochastic simulations. This technique has been widely employed in multiple gaming scenarios and achieved tremendous success, such as AlphaGo and AlphaZero [64] for computer Go Game. The basic MCTS algorithm involves iteratively search process with four steps for building a search tree:

(1) Selection. The agent, starting from an empty tree's root node, traverses the search tree's visited nodes and selects the next node according to the given selection strategy until the scalable node or leaf node is reached.

(2) Expansion. If this algorithm arrives at an expandable node, it expands the search tree by selecting an unvisited child node.

(3) Simulation. After finishing the expansion, if the current node is in a non-terminal state, the algorithm will conduct one or multiple independent simulations from the current node until it reaches the terminal state. In this process, the actions are chosen at random.

(4) Backpropagation. The node statistics on the path from the current node to the root are updated based on the search results. Note that the scores assessed are based on the termination state achieved.

To trade off the less tested paths with the best strategy identified so far, MCTS maintains a proper balance between exploration and exploitation by maximizing the Upper Confidence Bounds for Trees (UCT) when a child node $k$ is selected as follows, $\text{UCT} = \overline{X}_k + 2C_p\sqrt{\frac{2\ln n}{n_k}}$: where the first term, $\overline{X}_k$, is the average reward form arm $k$ and this term encourages the exploitation of higher-reward choices. It is generally understood that $\overline{X}_k$ to be within [0, 1]. In the second exploration term, $C_p > 0$ is a constant to satisfy the Hoeffding inequality with rewards in the range of [0, 1] [26]. $n$ is the number of times the current node has been visited and $n_k$ is the number of times child $k$ has been

visited. Generally, $n_k = 0$ produces a UCT value of $\infty$, so that all children of a node have a non-zero probability and are considered.

### G.2   LLM Reasoning with Monte Carlo Tree Search

LLMs have been invented, used in the past for autoregressive text generation, and are now very great at reasoning. Reasoning algorithms include prompt-based chain-of-thought (CoT) [22], planning-based represented by tree-of-thought (ToT) [24], which successfully achieved the LLMs' reasoning performance improvement. ToT combines the power of tree search (e.g., depth/breadth-first search) as an algorithm and LLMs' power as a heuristic to tradeoff evaluation and generation. Reasoning via Planning (RAP) [59] with Monto Carlo Tree Search (MCTS) performs reasoning exploration and obtains reward reasoning paths.

Recent studies [60] present that Monte Carlo Tree Search (MCTS) agents benefit from task-specific extension and expansion of the research tree. Specifically, the MCTS agents provide appropriate selection strategies for the state of the visit to guide the upcoming search based on the evaluation results (e.g., rewards and number of times the node has been visited) produced by the rollout and backpropagation process. Its mechanism coordinates exploration and thought exploitation within search space, which is superior to traditional depth-first search (DFS) or breadth-first search (BFS) algorithms based on the Tree of Thought (ToT). Building on MCTS, some studies have also explored the ability of the reasoning agent to provide search guidance. In catalyst design, [65] proposed Monte Carlo Thought Search, using LLM for complex scientific reasoning queries. [59] presents Reasoning via Planning (RAP), which adopts MCTS as a planning algorithm and repurposes the LLM as both a world model and a reasoning agent. Others like [66] experiment using the value function, a byproduct of the Proximal Policy Optimization (PPO) process, to guide the token-level decoding based on MCTS. In general, these approaches improve LLM's reasoning ability, whereas their performance on some challenging science tasks remains unsatisfying. In addition, this series of methods differs from our contribution, where we propose a value model approach as reward functions for optimizing the reasoning path and improving model output.

## H   Limitations

In this section, we discuss some limitations of the ReST-MCTS*.

**Generalization to other tasks, especially those without labels.** Similar to many existing self-training works, ReST-MCTS* also relies on ground-truth oracle labels in a supervised dataset to filter the responses in the first place; in the future, we need to show ReST-MCTS* can generalize to other reasoning tasks outside of math (like coding, agent, conversation, etc); in addition, for those very complicated tasks that require multistep planning and reasoning (like implementing the whole software like SWE-Agent), which does not have ground-truth answers, we need to propose a better way to collect reward feedback (from few human labeling and symbolic execution or solver), and train a generalizable reward model that can work and help for a wider range of tasks.

**Scale and diversity of proposed value model.** Although we trained a value model based on Mistral-7B: MetaMATH that performs better than the most advanced value model MATH-SHEPHERD, a larger scale value model backbone is still needed for better PRM training. In addition, the initial training set of the training proposal PRM was generated by SciGLM, a model that focuses on mathematical and scientific reasoning tasks but still lacks generality. While the current PRM achieves the best results on multiple mathematical and scientific reasoning tasks, such as MATH and SciBench, it's worth exploring more diverse training sets to expand into various fields in the future, such as code generation and agent planning.

**Self-training data filtering techniques.** As we mentioned in Section 1, the quality of reasoning trajectory affects the effectiveness of self-training, and generating a high-quality training set plays an important role. Therefore, we train the iterative process reward model to guide the tree search direction to obtain high-quality trajectories. On the other hand, since well-trained value models can help filter out the top-k generated trajectories with the highest process values, we also expect that a stronger and larger LLM model as the backbone of the value model might help to gain more.

# I    Broader Impact

ReST-MCTS* aims to introduce a general self-training approach that uses MCTS* to automatically label and generate process rewards, which will help generate high-quality datasets and improve the reasoning capabilities of LLMs. Fine-tuning a variety of LLMs on synthesized high-quality datasets can directly improve the performance of value models and generators and help to avoid the cost of manually generating process rewards during the training process reward model. The disadvantage is that a single reward model cannot be scaled to multiple domains, and we can solve this problem by training various reward models together on various reasoning domains. We believe that on the whole, the advantages outweigh the disadvantages.

# J    Reproducibility

We have made significant efforts to ensure the reproducibility of our all experimental results. The training code, tree search algorithm, and the evaluation details for the ReST-MCTS* are public in our repository.

**Training.** Detailed training information about the value model, self-training backbones, and experimental settings can be found in Section 4.1.

**Tree Search Algorithm.** Regarding the enhanced tree search algorithm MCTS*, please refer to the Algorithm 1 and public code.

**Evaluation.** We organized all evaluations, including the iterative self-training and value model, a variety of value models, performance comparison under the same search budget, and different reasoning policies. All details can be found in Section 4.2 for self-improvement evaluation and Section 4.3 for value models and reasoning policies comparison.

