# OpenReview forum: "ReST-MCTS*: LLM Self-Training via Process Reward Guided Tree Search"
_NeurIPS.cc/2024/Conference — NeurIPS 2024 poster_

### Official Review · Reviewer_S8hx · 2024-06-20

**Soundness:** 2
**Presentation:** 3
**Contribution:** 2
**Rating:** 5
**Confidence:** 5

**Summary:**

This paper proposed ReST-MCTS* to assist the large language model to answer reasoning questions. A variant of MCTS, which utilizes the evaluation of current state as the value function, is employed to
automatically annotate the process reward of each intermediate node via sufficient times of rollouts. A self-refine process is employed to finetune the LLMs. Tested on SciBench, ReST-MCTS* has achieved better performance than existing self-training approaches and reasoning policies.

**Strengths:**

1. This paper builds a value and policy network to assist the reasoning process of LLM , borrowing from the MuZero framework, and has achieved siginicant improvements.

2. Theoratical analysis is given to help the understanding of ReST-MCTS* algorithm.

3. The experiment is conducted comprehensively and convincingly.

4. The paper is written concisely and clearly.

**Weaknesses:**

1. More discussion about the value function is needed. In refincement learning, value is the expected total reward, rather than the evaluation of the current state. These two definitions are completely different, and I would like to see more explanations. For example:

    a) Can UCT still converge to the optimal solution when the simulation times approach infinity?

    b) In most cases, $v_k=v_{k-1}+w_{s_k}$ is established for two neighboring nodes in the search tree. Assume $S^k=\\{s_1^k,s_2^k,\cdots, s_N^k\\}$ denotes all children nodes of $s_{k-1}$. All nodes in $S^k$ share the same $v_{k-1}$ value in Equation (1). $v_{k-1}$ and $m_k$ are also the same in Equation (2), only $r_{s_k}$ is different. Considering that MCTS select the traversed node from the set of nodes with the same parent node while do simulation, why not predict $r_{s_k}$ directly?

    c) In the value backpropagation, $v_C$ of $s_t$ seems to be the weighted sum of the estimated values of all expanded nodes in the subtree rooted with $s_t$. What's the meaning of the average of state evaluation values? A larger evaluation value does not necessarily indicate a better expansion direction, especially when the distance to the goal is far.

    d) Using evaluation score as the value function makes the algorithm more likely to be a multi-step greedy algorithm, instead of a heuristic search algorithm like MCTS. What is your opinion on this matter?

2. Can you provide evidence of the quality of value function training based on prediction errors, in addition to the search results? Sometimes, even if the search results are good, it doesn't necessarily mean that the value function predictions are accurate. For example, set the estimated values to always be $0$, $A^*$ search can also perform well in some situations.

**Questions:**

1. Do different algorithms have differences in their actual running time, considering that MCTS requires a significant amount of Monte Carlo simulations to estimate action values, often resulting in higher computational requirements.

2. The definition of $r_{s_k}$ needs to be provided in advance. It is mentioned in Line 135, but is defined in Line 181. If I do not know that $r_{s_k}=1-r_{s_k}^{HE}$, observation 2 will be confusing to me.

3. Why is the assumption made that $w_{s_k}\in [0,1]$ when proving Theorem 1 in Appendix C.1? It is obvious that $w_{s_k}$ can be negative.

4. In line 648-650, $v_{k-1}\in [0,1]$ and $v_k\in [v_{k-1}, 1+v_{k-1}]$, why this makes $v_k$ in the range of $[0,1]$. It seems $v_k$ is in $[0,2]$.

**Limitations:**

Please refer to the Weaknesses and Questions part.

---

> ### Author Rebuttal · Authors · 2024-08-06
>
> ​​Thank you for acknowledging our contribution to LLM reasoning and raising valuable concerns and questions about various aspects of our work. We appreciate the time and effort you have dedicated to thoroughly assessing our work. To address your concerns and questions, we now provide a detailed response to each of them.
> ```
> Q2 & Q3 & Q4: Questions about specific definitions, assumptions, and conclusions.
> ```
> We want to clarify these questions first to help better understand this work. For Q2, we will surely rearrange the sequence of definitions as suggested. For Q3 and Q4 which focus on the deduction details in Appendix B.1, we sincerely apologize for making a mistake. In fact, reaching the conclusion of $v_k \in [0,1]$ does not require the assumption of $w_{s_k} \in [0,1]$. As an important factor measuring the correctness and contribution of a reasoning step, $w_{s_k}$ is designed to be signed and bounded within the range $[-1,1]$, rather than $[0,1]$. We’ve demonstrated the details in the **Official Comment**.
>
> ```
> W1: Concerns on the definition of quality value.
> ```
> We appreciate your concerns about the design of weighted reward and quality value. They remind us that there are a few things we may have neglected and require more explanation. Please refer to the **Global Author Rebuttal** for a general explanation of our approach design.
>
> a) **Your concerns regarding the convergence of UCT for our case are profound and thoughtful.** As mentioned in the **Global Author Rebuttal**, our methodology is offline RL and doesn’t face issues with UCT convergence. However, we agree that the convergence of UCT in our setting for online RL is unclear. In consideration that our definitions of reward and value are novel, it certainly requires considerable effort to design an appropriate online RL paradigm that aligns with these definitions.
>
> b) In fact, under our definition, the reasoning distance $m_k$ is the minimum reasoning steps required to access the final answer starting from a node with partial solution $p_k=[s_1, s_2, \cdots, s_k]$. This means when we generate children nodes of $s_{k-1}$, the reasoning distance of these nodes, denoted as $m_k^i(i=1, 2, \cdots, N)$, is already different. Different children make different contributions to the final answer, leading to varied search directions that require different numbers of reasoning steps. Thus, **though $v_{k-1}$ is the same, both $r_{s_k}$ and $m_k$ are different**.
>
> c) The value backpropagation process is not compulsory. However, we regard this process as a heuristic that provides insight into the selection of search direction. Updating value estimation using the average method promotes exploitation in more promising directions in general.  Please see the details in our **Official Comment**.
>
> d) We believe former discussions have addressed this question. By adopting UCT and value backpropagation, our algorithm values exploration as important as exploitation. Concerning implementation, the exploration constant also allows the different extents of the exploration. Though we use a different value definition, the core idea of MCTS has not changed.
>
> ```
> W2: Concerns on the quality of the value model.
> ```
> We appreciate your concerns that our value model may not be truly effective. To address your concern, we present two pieces of evidence aside from the results of the search.
>
> Firstly, our value model achieves an accuracy of approximately 70% with an absolute tolerance of 0.1 on the test set consisting of 14k data samples, as demonstrated in Appendix D.1. This means the value model can already gain considerable knowledge even before self-training, justifying its quality.
>
> Moreover, we also experiment to compare the performances of critics alone. We use the same policy and the same CoT sampling strategy to generate solutions for GSM8K and MATH500. These solutions are then evaluated and filtered using different methods. Results shown in Table 3 indicate that the filtration method based on our value model + SC achieves the best accuracy on both datasets, outperforming baselines like SC, ORM, and PRM of Math-Shepherd. This also justifies the quality of our value model, since all other influences are eliminated.
>
> ```
> Q1: Questions on running time and computational requirements of different algorithms.
> ```
> Concerning computational costs, we have already compared the number of tokens consumed by each algorithm to achieve certain accuracy on MATH and SciBench, as shown in Figure 2. Results reveal that to reach a certain expected accuracy, MCTS* basically requires fewer tokens than other algorithms like SC and ORM+BoN. This means that based on the same expected standards, MCTS$^*$ outperforms other algorithms while maintaining a reasonable computation cost. As for the actual running time, we have recorded the average running time of different algorithms (under our basic experiment settings) on a single question, as shown below.
>
> | Method | CoT+SC | ORM+BoN | PRM+BoN | MCTS* |
>  --- | --- | --- | ---| ---
> | Running time (s) | 41 | 43 | 73 | 108 |
> | Accuracy on MATH (%) | 37.0 | 33.5 | 34.5 | 41.5 |
>
> Indeed, MCTS$^*$ spends more time on exploration and simulation than other simple algorithms. However, since our method adopts a different design of value, it doesn’t require massive Monte Carlo estimations. This reduces the running time of our algorithm and limits the time consumption to a reasonable range. Notice that MCTS* can achieve high accuracy that other algorithms can never attain even at unlimited cost, we believe this extra time is fairly acceptable.
>
> Lastly, we kindly appreciate your in-depth evaluation of our work and thank you for your valuable questions and suggestions, which have greatly contributed to improving our work. If you believe that our responses have satisfactorily addressed your concerns about the issues, we kindly request that you consider adjusting the final evaluation to reflect this.

---

> > ### Author Response · Authors · 2024-08-10
> > **Looking forward to your feedback**
> >
> > Dear reviewer S8hx,
> >
> > thank you very much for your valuable feedback. We hope that our responses and clarifications have addressed your questions and concerns.  If you believe that our responses have satisfactorily addressed your concerns, we kindly request that you consider adjusting the final rating to reflect this.
> >
> > If there are any remaining concerns or require additional clarification, please let us know. We are looking forward to your reply. Thank you for your time and efforts on this paper.
> >
> > Best regards,
> >
> > Authors of ReST-MCTS*

---

> > > ### Comment · Reviewer_S8hx · 2024-08-12
> > >
> > > Thank you for your response. In my opinion, using 'evaluation' as a reward does not align with the optimization objectives of reinforcement learning. The objective is to optimize the evaluation of the final output, instead of evaluations of all states in the generation trajectory, although sometimes there is a certain relationship between the two. Further explanations or experiments are needed to explain why this definition is chosen over others, such as using the improvement in evaluation of adjacent states as the reward. Therefore, I maintain my score.

---

> ### Author Response · Authors · 2024-08-06
> **More details related to value backpropagation process and deduction.**
>
> Thank you for your valuable questions and thorough evaluation of our work! Here we present some details related to your concerns and questions, as mentioned in our rebuttal.
>
> ```
> W1: Concerns on the question about the value backpropagation process.
> ```
>
> We regard the value backpropagation process as a heuristic that provides insight into the selection of search direction. Since the quality value reflects the valid progress made toward a correct final answer, a higher value generally represents that the corresponding direction is more accessible to the answer, more probable to be the right direction, and more promising for the policy to make progress. Therefore, updating value estimation using the average method promotes exploitation in more promising directions in general.
>
> On the other hand, this doesn’t mean our algorithm is prone to ignore directions that currently receive low-value estimation. As we utilize UCT-based selection criteria, nodes that are overly exploited will not be selected in future rollouts. Besides, the average value method also helps to adjust inaccurate value estimation. If a node’s value is underestimated, in future rollouts its children nodes will still be evaluated. If the children nodes reveal some promising directions, the update of value will adjust the underestimation, and vice versa.
>
>
> ```
> Q2 & Q3 & Q4: Concerns on the problems with deduction.
> ```
> We want to present the correct deduction here. For quality value, we can derive the expected boundedness from Inequality 17 in our paper, because:
>
> $w_{s_k} \leq |1-v_{k-1}|$, $v_k=max(v_{k-1}+w_{s_k},0)$ and $v_0=0$.
>
> Inductively, we can derive that $v_k \in [0,1]$ as long as $v_{k-1} \in [0,1]$, eventually reaching the conclusion that for every k, $v_k \in [0,1]$.
>
> We will modify these mistakes in the revised manuscript. Thank you for such a meticulous inspection of our paper!

---

> ### Author Response · Authors · 2024-08-12
> **Thank you for your reply! We have addressed the concerns on the 'evaluation' as a reward.**
>
> Dear Reviewer S8hx,
>
> thank you for your comments! Regarding the ‘evaluation’ as a reward, we would like to further explain and justify our approach from the following five aspects.
>
> (1) We’d like to highlight that **we predominantly focus on complex reasoning scenarios. In fact, for this scenario, the evaluation of intermediate states is very important.** Unlike most traditional RL tasks where the final reward is regarded as more important, complex reasoning processes rely more on intermediate steps, making the quality of an intermediate step non-negligible.
>
> (2) Methods based on traditional RL adopt sparse reward, where a reward signal is received only when the reasoning is finished. However, these methods neglect an important fact: Even when a reasoning trace looks good overall (concerning the final outcome), it may still suffer from intrinsic logical faults [1]. This point has been mentioned from line 32 to line 37 in our submission. Similarly, **although traditional ways of modeling the reward are relatively easy and concise, omitting intermediate rewards makes it a compromise strategy.**
>
> (3) To achieve more complex reasoning, **modeling intrinsic rewards may be inevitable, this elevates the significance of Process Reward Models (PRMs).** However, the reward for intermediate reasoning steps is hard to model. A reward should reflect the transition of states and the value of an action. Since the transition of states is determinant once an action is taken when reasoning, the main problem comes to the second term, which is somewhat unclear for reasoning scenarios. To tackle this issue, we referred to the common process of exam grading. In this process, graders examine the contribution of each solution step, assigning a total score (reward) based on the cumulated score of each step, this analogizes our design of the reasoning process and reward. **In our approach, we not only optimize the policy to attain a higher final reward but also encourage it to explore and generate better intermediate steps through our careful design of reward and algorithm.** A policy trained with our method learns to seek more promising search directions (high intermediate rewards) and will look for other alternatives when it cannot make more progress in the current direction, just like a human test taker in a math test.
>
> (4) Furthermore, **the design of PRM reward/value of our work thoughtfully ensures accurate estimation of quality of actions.** In our definition, we incorporate the process reward $r_{s_k}$ (evaluates the probability of correctness of a single step) and the reasoning distance $m_k$ (evaluates contribution or importance, though in an indirect way). These factors involve information that can help evaluate a step more accurately, thus we believe involving these factors in the design of reward and value may improve the effectiveness of our method. Under this setting, weighted reward reflects correctness and contribution, while quality value reflects the progress made in the right direction. An action (step) receives a higher reward when it correctly tackles more components of the problem, which is natural and reasonable.
>
> (5) Finally, as shown in Figure 2 and Table 3, experiment results indicate that **our designed PRM outperforms both traditional PRM and ORM in various aspects.** This justifies the validity of our design.
>
> Reference:
>
>  [1]  T. Lanham, A. Chen, A. Radhakrishnan, B. Steiner, C. Denison, D. Hernandez, D. Li, E. Durmus, E. Hubinger, J. Kernion, et al. Measuring faithfulness in chain-of-thought reasoning. https://arxiv.org/abs/2307.13702
>
> Once again, we sincerely thank you for your thoughtful questions, which have greatly contributed to improving our work. We believe that the revisions we have made adequately address your concerns and questions regarding the ‘evaluation’ as a reward of ReST-MCTS*. If you believe that our responses have satisfactorily addressed your concerns about the issues, we kindly request that you consider adjusting the final evaluation to reflect this.

---

> > ### Author Response · Authors · 2024-08-14
> > **Looking forward to your feedback**
> >
> > Dear Reviewer S8hx,
> >
> > Thank you very much for your valuable comments. As we approach the 'evaluation' as a reward of the discussion phase, we would like to know whether the responses have addressed your concerns about the significance of our work and the reward design issue.
> >
> > If you believe that our responses have satisfactorily addressed your concerns, we kindly request that you consider adjusting the final rating to reflect this.
> >
> > If there are any remaining concerns, please let us know. We are more than willing to engage in further discussion and strive to address any remaining issues to the best of our abilities. We are looking forward to your reply. Thank you for your time and efforts on this paper.
> >
> > Best regards,
> >
> > Authors of ReST-MCTS*

---

### Official Review · Reviewer_r9nL · 2024-07-13

**Soundness:** 3
**Presentation:** 2
**Contribution:** 2
**Rating:** 5
**Confidence:** 3

**Summary:**

This paper proposes a novel approach for self-training large language models (LLMs) that combines process reward guidance with Monte Carlo Tree Search (MCTS). This method generates high-quality reasoning traces and per-step values to train policy and reward models, eliminating the need for manual annotation. Experimental validation on multiple benchmarks shows that ReST-MCTS* outperforms existing self-training methods by achieving higher accuracy in reasoning tasks.

**Strengths:**

- The combination of Monte Carlo Tree Search (MCTS) with process reward models represents a novel method for improving self-training in LLMs. This integration allows for automatic generation of high-quality reasoning traces, which is a significant advancement over existing methods.
- The paper provides clear definitions and theoretical support for key concepts such as quality value and weighted reward. This enhances its robustness and effectiveness in the self-training process for reasoning problem.

**Weaknesses:**

- This paper introduces a method to evaluate values for each intermediate step in the reasoning process. Providing more evidence to demonstrate the validity and reasonableness of these intermediate values would make the paper more convincing.
- Scalability. While the method aims to eliminate manual annotation, the scalability of the proposed approach in extremely large datasets or more complex reasoning tasks might still be a challenge. Maybe complex task require more intermediate steps and wonder if proposed quality value still work for those cases. Providing additional strategies or evidence to support scalability would strengthen the paper.
- Although the paper demonstrates improved performance on selected benchmarks, a broader range of datasets and tasks would provide a more comprehensive validation of the method’s generalizability and robustness.

**Questions:**

- Typically, simultaneously training the reward and policy can lead to instability and convergence difficulties. Do you have any concerns about this issue, and how do you address it? Are there any techniques you employ to mitigate these challenges?

- The notation is confused. V_{\theta} in line 149 is value function but in line 175 is process reward model.

**Limitations:**

The experiments were conducted on a restricted set of benchmarks, which limits the generalizability of the findings.

As mentioned in the weaknesses, the scalability of the proposed approach in handling extremely large datasets or more complex reasoning tasks remains uncertain. Addressing this issue with additional strategies or evidence would strengthen the paper and its applicability.

---

> ### Author Rebuttal · Authors · 2024-08-06
>
> Thank you for acknowledging our contribution to LLM self-training, clear definitions, and theoretical support and raising valuable concerns and questions about various aspects of our work. We appreciate your dedicated time and effort to thoroughly assess our work. We provide a detailed response to each of them to address your concerns and questions.
>
> ```
> W1: Concerns on validity and reasonableness of proposed reward/value design in ReST-MCTS*.
> ```
> We appreciate the reviewer's attention to this concern. Allow us to delve deeper into this concern. The formulation of our weighted reward and quality value (as depicted in Equations 1 and 2) is structured with a specific rationale in mind.
>
> In our methodology, **we assign varying rewards to different reasoning steps based on their contributions**. This contrasts with previous approaches like Math-Shepherd, which often rely on sparse rewards and the probability of success as values, neglecting the nuanced importance of each step and potentially leading to suboptimal search outcomes.
>
> Our approach integrates the concept of process reward $r_{s_k}$ (assessing the probability of correctness) and reasoning distance $m_k$ (evaluating contribution or importance indirectly). By incorporating these factors into our reward and value design, we aim to enhance the accuracy of step evaluation. The effectiveness of ReST-MCTS$^*$ over MS, as demonstrated in Table 3, underscores the practical benefits and advantages of our design.
>
> Furthermore, through extensive experiments detailed in Table 2, we showcase the efficacy of our design across various tasks and LLMs, thereby verifying the scalability and robustness of our method.
>
> ```
> W2 & L2: Concerns on scalability.
> ```
> While this article mainly addresses mathematical reasoning tasks and shows effectiveness, it is still suitable for working with very large data sets or more complex tasks, such as code generation scenarios, which require additional intermediate steps due to the need for them. To enhance scalability, additional strategies can be explored. For instance,
>
> (1) for longer and more complex codes, higher-level structures can be used as a single reasoning step. These structures could include lines of code, code blocks, or even entire functions.
>
> (2) similar to approaches used in math reasoning tasks, a similar PRM provides expert reward feedback for value function estimation. In this way, the MCTS algorithm can always filter out better traces regardless of the policy’s competence. Alternatively, if we continue to rely on MC simulations, the policy and an existing reward model could be updated during this process. RL methods such as Q-learning can be adopted.
>
> ```
> W3 & L1: Concerns on a broader range of datasets and tasks.
> ```
> To further bolster the method's generalizability and robustness, expanding the evaluation to encompass a broader range of datasets and tasks beyond common mathematical reasoning could offer a more comprehensive validation.
>
> As highlighted previously, this study primarily concentrates on mathematical reasoning tasks, which is reflected in the evaluation of mathematical benchmarks such as MATH (Table 2, Figure 2), GSM8K (Table 3), MATH500 (Table 3), and various Math subsets (Table 4).
>
> Moreover, the method's performance is also assessed on scientific reasoning benchmarks, including GPQA_{Diamond} and CEval-Hard (Table 2), as well as SciBench datasets covering Mathematics, Chemistry, and Physics (Table 4, Table 6, Figure 5), and SciEval (Table 7).
>
> Corresponding to the W2 & L2, this paper provides a more comprehensive assessment of the method's effectiveness across a wider spectrum of applications and facilitates a deeper understanding of its generalizability.
>
> ```
> Q1: Concerns about simultaneously training the reward and policy.
> ```
> We acknowledge the importance of considering the issue of online RL. However, **it is vital to emphasize that our methodology primarily focuses on the offline self-training paradigm**. Within this framework, data is generated via MCTS$^*$ using a static policy and critic within each iteration. The newly synthesized data is verified and filtered according to the ground truth rather than the outputs of the value model. Subsequently, this filtered data is utilized to individually train the policy and value models, aligning our algorithm with an offline and relatively stable approach.
>
> In tackling the instability and convergence challenges inherent in simultaneously training the reward and policy and enhancing the overall performance and robustness of the training process, typical of online RL strategies, we can employ several techniques:
>
> Experience Replay: firstly, during training, the agent stores transition in a replay buffer, accumulating experiences over time. Then, instead of using experiences immediately, the agent samples mini-batches of experiences randomly from the replay buffer during training. By sampling randomly from past experiences, experience replay breaks the temporal correlations present in sequential data, which can help prevent the model from getting biased toward recent experiences. Finally, using experience replay can lead to more stable and efficient learning by providing a diverse set of experiences for the agent to learn from, smoothing out the learning process.
>
> ```
> Q2: Concerns on the $V_\theta$.
> ```
> Thanks for the suggestion, $V_{\theta}$ is indeed the definition of the process reward model. We will update this typo (line 149) in our manuscript.
>
> Once again, we sincerely thank you for your thoughtful evaluation and valuable suggestions, which have greatly contributed to improving our work. We believe that the revisions we have made adequately address your concerns and questions regarding quality value, scalability, and related aspects of ReST-MCTS*. If you believe that our responses have satisfactorily addressed your concerns about the issues, we kindly request that you consider adjusting the final evaluation to reflect this.

---

> > ### Author Response · Authors · 2024-08-10
> > **Looking forward to your feedback.**
> >
> > Dear Reviewer r9nL,
> >
> > Thank you very much for your valuable comments. As we approach the conclusion of the rebuttal phase, we would like to know whether the responses have addressed your concerns about the significance of reward/value design, the scalability issue, and evaluation benchmarks.
> >
> > If you believe that our responses have satisfactorily addressed your concerns, we kindly request that you consider adjusting the final rating to reflect this. Other reviewers have acknowledged solid theoretical foundations and comprehensive experiments and gave positive evaluations. (One Reviewer has raised the rating from 5 to 7. )
> >
> > If there are any remaining concerns that have led to a negative evaluation, please let us know. We are more than willing to engage in further discussion and strive to address any remaining issues to the best of our abilities. We are looking forward to your reply. Thank you for your time and efforts on this paper.
> >
> > Best regards,
> >
> > Authors of ReST-MCTS*

---

> > > ### Comment · Reviewer_r9nL · 2024-08-11
> > >
> > > Thank you for answering my questions. Your responses have resolved some of my concerns. After reconsidering the contributions of this paper, I believe it would be good to raise the score from 4 to 5.

---

> > > > ### Author Response · Authors · 2024-08-12
> > > > **Thanks for your great reviews!**
> > > >
> > > > Dear Reviewer r9nL,
> > > >
> > > > Thank you very much for your response and for recognizing our contributions. You mentioned that our responses addressed some of your concerns, and we are eager to tackle the remaining ones and engage in further discussion. If possible, could you kindly share the remaining concerns with us? We are looking forward to your reply and appreciate the time and effort you have dedicated to reviewing our paper.
> > > >
> > > > Best regards,
> > > >
> > > > Authors of ReST-MCTS*

---

### Official Review · Reviewer_rDpc · 2024-07-14

**Soundness:** 3
**Presentation:** 3
**Contribution:** 3
**Rating:** 7
**Confidence:** 4

**Summary:**

This paper introduces a novel approach for self-training large language models (LLMs) called ReST-MCTS*. This method integrates process reward guidance with Monte Carlo Tree Search (MCTS) to collect high-quality reasoning traces. These traces are then used to train policy and reward models without relying on manual annotations for every reasoning step. The paper claims that this method outperforms existing self-training techniques in terms of accuracy on several reasoning tasks.

**Strengths:**

1. The application of MCTS to improve the capabilities of LLMs is a highly promising approach. This integration allows for more structured and effective exploration of reasoning paths, leading to better model performance.
2. The proposed method has demonstrated its effectiveness across various reasoning tasks, significantly enhancing the performance of LLMs.

**Weaknesses:**

1. One major issue with the paper is the lack of novelty, particularly due to the absence of comparison with AlphaLLM [1]. AlphaLLM also uses MCTS to enhance LLM performance, and the ideas and implementation in both papers are strikingly similar. The omission of a comparison or even a citation significantly undermines the contribution and novelty of this paper.
2. The method still treats each step's reward equally concerning the final answer, only decreasing based on the distance from the root node. This approach does not effectively differentiate the importance of various steps, failing to allocate different weights to different steps appropriately.

Reference:
[1]. Toward Self-Improvement of LLMs via Imagination, Searching, and Criticizing. https://arxiv.org/abs/2404.12253

**Questions:**

1. How does the method avoid the scenario where an incorrect intermediate step leads to the correct final answer as mentioned in the paper? Can the Process Reward Model (PRM) alone handle this issue? If an incorrect step during the expansion phase eventually leads to the correct result, would this negatively impact the value function training?
2. Why is the weighted value designed in its current form? Is there any theoretical justification or practical benefits of this design?
3. In the LLaMA-3-8B-Instruct results, is the performance of ReSTEM in the first iteration (3.84) a typographical error? Why is the performance so low?

**Limitations:**

1. Lack of comparison or missing citation of important baseline.
2. Some design details and implementations can be explained in more detail.

---

> ### Author Rebuttal · Authors · 2024-08-06
>
> Thank you for your valuable feedback and concerns regarding novelty, the design of reward/value/PRM, and related aspects of our paper. We genuinely appreciate your dedicated time and effort to thoroughly assess our work. We have carefully considered your comments and have made the necessary responses to address these concerns and improve the transparency and credibility of our research. Below, we provide a detailed response to each of your points.
>
> ```
> W1 & L1: Concerns on lack of comparison with AlphaLLM.
> ```
> **The concurrent work AlphaLLM appeared on ArXiv on April 18th.** We appreciate you bringing this to our attention and will cite AlphaLLM in our paper.
>
> As an approach that aims to enhance LLM inference, AlphaLLM utilizes a tailored MCTS algorithm and critic models to provide precise feedback. Even though AlphaLLM also adopts MCTS and critic models for self-improvement, their approach is different from ours in various crucial aspects, as elaborated below.
>
> **1) Design of MCTS algorithm.**
> For the level of search, AlphaLLM’s $\eta$MCTS considers options as action, with termination signals delivered by a termination function $\beta$. In contrast, we use reasoning steps as action, which is achieved through tailored prompt design.
>
> Concerning critic models, we use a single value model to provide evaluation for intermediate nodes. The model is trained to predict specially designed quality values that reflect completeness and correctness of partial solutions, rather than estimating the conventional definition of value function in RL.
>
> In addition, we also incorporate self-critic mechanisms into the tree search algorithm to provide insights for the policy (Appendix C.1), which AlphaLLM does not adopt.
>
> **2) Definition of reward/value.**
> Our definition of weighted reward and quality value is novel, leading to significant differences between our method and AlphaLLM across various processes such as critic model training, data synthesizing, and data filtration. Since our design of quality value involves information on process reward and reasoning distance, our value model trained on this target can naturally provide sufficient feedback during the search, with no need for implementing other critic models mentioned by AlphaLLM.
>
> **3) Self-Training algorithm.**
> Although AlphaLLM also includes iterative self-training, the implementation method varies greatly. Most importantly, their critical model is static throughout the iterations, which means they focus more on the improvement of policy. In comparison, we also consider the impacts of self-training on the critic value model.
>
> As demonstrated in Algorithm 1, we calculate process rewards and quality values according to the final search tree of questions within each iteration, which are then used as new training data for the value model.
>
> ```
> W2 & Q2 & L2: Concerns and questions on design of weighted reward/quality value.
> ```
> We appreciate the reviewer's concern and query about the validity and effectiveness of our reward/value design. Please allow us to elaborate more on this issue. We design the weighted reward and quality value this way (Equation 1, 2) for two main reasons. Please see our **Official Comment** for more details on this part.
>
> 1) First, we believe that **different reasoning steps should receive varied rewards according to the contribution they make.** To achieve this, we incorporate the process reward $r_{s_k}$ (evaluates the probability of correctness) and the reasoning distance $m_k$ (evaluates contribution or importance, though in an indirect way). These factors involve information that can help evaluate a step more accurately, improving the effectiveness of our method.
>
> W2: Actually, each step’s reward is equal only when the corresponding reasoning trace is the idealized “perfect” solution, where all steps are important, necessary, and concise. Otherwise, steps that make more contributions to the solution have a smaller reasoning distance and higher quality value.
>
> 2) Another important reason is scalability and accessibility. Unlike some methods, ours can easily obtain an estimation of each node’s reasoning distance as illustrated in Section 3.2. This enables scaling up our self-training process, which is a significant advantage and practical benefit of our design. Furthermore, Table 2 proves that our design is indeed effective among various tasks and LLMs, further verifying the scalability of our method. We believe this answers the reviewer’s Q2.
>
> ```
> Q1: Concerns on the impact of incorrect intermediate steps.
> ```
> We agree that an incorrect intermediate step that reaches the correct answer will be harmful to the iterative training of the value model. However, the possibility of the occurrence of this issue in our work is very small. Our value model is firstly trained on a credible data set that instructs the model to distinguish correct intermediate steps from false ones before self-training. Therefore, during self-training, the MCTS* algorithm can avoid expanding false nodes naturally by probability.
>
> Another measure for this issue is to filter out the nodes on a trace that reach the correct answer but obtain lower quality values than their parent node (this means they have a negative reward). This helps to purify the newly generated train data based on previous knowledge of the PRM itself.
>
> ```
> Q3: Question on the low performance of LLaMA-3-8B-Instruct.
> ```
> Thank you for raising this question. We have thoroughly reviewed and re-evaluated the results multiple times to ensure the accuracy and reliability of the final result. The result should be 30.84 and we will update this typo.
>
> Once again, we sincerely thank you for your thoughtful evaluation and valuable suggestions, which have greatly contributed to improving our work. If you believe that our responses have satisfactorily addressed your concerns about the issues, we kindly request that you consider adjusting the final evaluation to reflect this.

---

> ### Author Response · Authors · 2024-08-06
> **More details related to the design of weighted value.**
>
> Thank you for your valuable questions and thorough evaluation of our work! Here we present some details related to your concerns and questions, as mentioned in our rebuttal.
> ```
> W2 & Q2 & L2: More details on the design of weighted reward/quality value.
> ```
> W2: Actually, our method only treats each step’s reward equally when the corresponding reasoning trace is the idealized “perfect” solution, where all steps are important, necessary, and concise. In this situation, it’s reasonable and natural to assign equal rewards. In other circumstances, suppose we perform expansion at a node with a partial solution $p_{k-1}$, then its children nodes will have different $r_{s_k}$ and $m_k$ according to the generated step $s_k$. For steps $s_k$ that make more contributions to the solution, their corresponding child node requires fewer steps to reach the correct answer. This means they have a smaller reasoning distance, resulting in higher weighted reward and quality value. Therefore, **our method can differentiate the importance of steps, as long as the critic model can estimate the designed reward accurately**.
>
> Q2: It may be more direct to think of modeling the “importance” or “contribution” of a step and train a critical model to predict that. However, it is extremely difficult to acquire or access sufficient data that allows training of a valid critic model, not to mention that the modeling of “importance” or “contribution” is already fairly hard. As an alternative, our definition of reasoning distance reflects these factors in an indirect way. Through the tree-search based data synthetic process, we can easily obtain an estimation of each node’s reasoning distance using the method illustrated in Section 3.2. This enables scaling up of our self-training process, which is a crucial goal of our work.

---

> ### Comment · Reviewer_rDpc · 2024-08-07
>
> Thanks for your response. My main concerns are addressed. Although I believe that AlphaLLM does not strictly qualify as concurrent work due to the significant time gap between its arXiv posting and the NeurIPS submission deadline, I hope that in subsequent versions, the authors can compare their method with AlphaLLM and further clarify the differences between the two approaches.
>
> I have raised my rating to 7. Congrats on your great work!

---

> ### Author Response · Authors · 2024-08-08
> **Thanks for your great reviews!**
>
> Thanks for your great reviews! In our final version, we will provide a thorough comparison with AlphaLLM. Thank you again for the score adjustment!

---

### Official Review · Reviewer_7P64 · 2024-07-29

**Soundness:** 3
**Presentation:** 2
**Contribution:** 3
**Rating:** 5
**Confidence:** 3

**Summary:**

The paper introduces ReST-MCTS*, a novel framework for self-training LLMs using MCTS combined with process reward guidance. The core innovation is in addressing the limitations of traditional self-training methods, which often include incorrect intermediate reasoning steps despite producing correct final answers. ReST-MCTS* leverages a modified MCTS algorithm, that integrates a process reward model to estimate the probability that each intermediate step contributes to the correct final answer. This allows for the automatic generation of high-quality reasoning traces without requiring dense human annotations. The inferred rewards serve as value targets for refining the process reward model and selecting high-quality traces for self-training the policy model. Experimental results on benchmarks like SciBench and MATH show that ReST-MCTS* not only outperforms previous self-training approaches (e.g., ReSTEM, Self-Rewarding LM) but also enhances the LLMs' accuracy through iterative self-improvement.

**Strengths:**

- Originality:
  - Proposes an innovative integration of MCTS with process reward guidance.
- Quality:
  - Theoretical foundations are robust, and the methodology is supported by extensive experimental validation.
  - Demonstrates substantial improvements in performance on multiple benchmarks, showcasing the effectiveness of the proposed approach.
- Significance:
  - The approach addresses a critical challenge in multi-iteration LLM self-training, enhancing the quality of generated training data.
ope

**Weaknesses:**

- Related Work: While this paper mentions related work, such as Xidong, et al.'s [1], it does not adequately discuss the differences between its handcrafted value target and TD(λ) in [1]. A more thorough comparison with above approach would provide better context and highlight the novel contributions of this work. (e.g. TS-LLM is missing in figure 6)
- Clarity: The paper suffers from clarity issues, as many definitions and notations are introduced later in the text rather than upfront. Additionally, several important figures are placed in the appendix, making it difficult for readers to follow the key points without constantly referring to supplementary materials.

[1] Feng, X., Wan, Z., Wen, M., Wen, Y., Zhang, W., & Wang, J. (2023). Alphazero-like tree-search can guide large language model decoding and training. arXiv preprint arXiv:2309.17179.

**Questions:**

- Other than the experiment in Figure 2 showing that multi-iteration ReST-MCTS* significantly outperforms SC, the other experiments only demonstrate marginal gains over SC. What do you think is the reason behind this?

**Limitations:**

NA (Good Enough)

---

> ### Author Rebuttal · Authors · 2024-08-06
>
> Thanks a lot for acknowledging the strengths of this work as an innovative self-training method, robust theoretical foundations, and extensive benchmarks. We have given a detailed discussion with related work, clarity issues, and the performance of SC.
>
> ```
> W1: Concerns on lack of comparison between proposed ReST-MCTS* and TS-LLM.
> ```
> In Table 1, this paper initially contrasts TS-LLM with our proposed ReST-MCTS$^*$ concerning reasoning policy and reward guidance. To offer more depth, we have conducted a comprehensive comparison between these two approaches, aiming to offer a richer context and underscore the unique contributions of our work.
>
> As an approach that aims to enhance LLM inference decoding and training, TS-LLM utilizes a tailored MCTS algorithm and value function to provide precise feedback. Their work proved that TS-LLM is indeed an efficient framework that significantly boosts the performance of LLMs without requiring extra data annotations. Even though TS-LLM also adopts MCTS and value function for self-improvement, their approach differentiates from ours in various crucial aspects elaborated as follows.
>
> **(1) Design of MCTS algorithm.**
> For the level of search, TS-LLM considers each sentence or each token as an action, with termination signals delivered by a termination function $\alpha$. In contrast, we use reasoning steps as action, which is achieved through tailored prompt design. Our proposed value model is trained to predict specially designed quality values that reflect completeness and correctness of partial solutions, rather than estimating the conventional definition of value function in RL. Additionally, we also incorporate self-critic mechanisms into the tree search algorithm to provide insights for the policy, as mentioned in Appendix C.1, which TS-LLM does not adopt.
>
> **(2) Definition of reward/value.**
> Our definition of weighted reward and quality value is novel, leading to significant differences between our method and TS-LLM across various processes such as critic model training, data synthesizing, and data filtration. Since our design of quality value involves information of $r_{s_k}$, $m_k$, and $v_{k-1}$, our value model trained on this target can naturally provide sufficient feedback during the search, with no need for implementing other critic models mentioned by TS-LLM.
>
> **(3) Self-Training algorithm.**
> Although TS-LLM also includes iterative self-training, the implementation method varies greatly. Most importantly, their reward model is ORM throughout the iterations, which means they focus more on the improvement of policy. In comparison, we consider the impacts of self-training on the critic value model. As demonstrated in Algorithm 1 and Appendix C.2, by carefully designing the data synthetic process, we calculate process rewards and quality values according to the final search tree of questions within each iteration, which are then used as new training data for the value model. Through comprehensive experiments, we further reveal that both the policy and the critic can be continuously improved for multiple iterations of self-training (Table 2), achieving significant enhancement in search accuracy under reasonable token consumption (Figure 2).
>
> In addition, we will add the key differences between TS-LLM and our work in Figure 6 in our revision.
>
> ```
> W2: Concerns on clarity issues.
> ```
> We sincerely apologize for the inadequate arrangement of the sequence of definitions. We will surely modify the sequence definitions introduced according to your suggestion to make our paper more intelligible to readers. Once again, we would like to express sincere apology for our negligence on these details, and we will certainly modify them in the revised manuscript. Thank you for such a meticulous inspection of our paper!
>
> ```
> Q1: Question about the effectiveness of SC.
> ```
> Regarding the performance comparison between SC and ReST-MCTS$^*$, we provide all the results among them. In fact, in addition to the experiments in Figure 2 showing that ReST-MCTS$^*$ with multiple iterations is significantly better than SC, other experiments also show that ReST-MCTS$^*$ has a significant improvement over SC. For instance,
>
> In Figure 2, ReST-MCTS$^*$ significantly outperforms SC after multi-iterations, which verifies the effectiveness of self-training algorithm.
>
> In Table 4, CoT is indeed the CoT-SC setting, we can find that ReST-MCTS$^*$ outperforms CoT-SC on the separate accuracy of most subjects and the final average accuracy.
>
> In Table 3, we first provide the results of ReST-MCTS$^*$ on GSM8K and MATH500, which are 86.8 and 37.4, respectively, and then calculate the relative improvement of value models (ORM, MS, and ReST-MCTS*) compared to SC.
>
> | Dataset | SC | ORM | \% Improv. |
>  --- | --- | --- | ---
> | GSM8K | 83.9 | 86.2 | 2.74 |
> | |  | MS | \% Improv. |
> | |  | 87.1 | 3.81 |
> | |  | ReST-MCTS* | \% Improv. |
> | |  | 86.8 | 3.46 |
>
>
> | Dataset | SC | ORM | \% Improv. |
>  --- | --- | --- | ---
> | MATH500 | 35.1 | 36.4 | 3.70 |
> | |  | MS | \% Improv. |
> | |  | 37.3 | 6.27 |
> | |  | ReST-MCTS* | \% Improv. |
> | |  | 37.4 | 6.55 |
>
> In the whole view, ReST-MCTS$^*$ significantly outperforms SC in multiple experimental settings, e.g., multi-iterations, scientific reasoning benchmark SciBench, and mathematical reasoning benchmarks.
>
> Once again, we sincerely thank you for your thoughtful evaluation and valuable suggestions, which have greatly contributed to improving our work. We believe that the revisions we have made adequately address your concerns and questions regarding the comparison and effectiveness of SC. If you believe that our responses have satisfactorily addressed your concerns about the issues, we kindly request that you consider adjusting the final evaluation to reflect this.

---

> ### Author Response · Authors · 2024-08-10
> **Looking forward to your feedback**
>
> Dear reviewer 7P64,
>
> thank you very much for your valuable feedback. We hope that our responses and clarifications have addressed your questions and concerns. If you believe that our responses have satisfactorily addressed your concerns, we kindly request that you consider adjusting the final rating to reflect this.
>
> If there are any remaining concerns or require additional clarification, please let us know. We are looking forward to your reply. Thank you for your time and efforts on this paper.
>
> Best regards,
>
> Authors of ReST-MCTS*

---

### Author Rebuttal · Authors · 2024-08-06

Dear ACs and reviewers,

thank you very much for your valuable feedback. We list the main issues raised by reviewers and explain them below.

```
The motivation and reason behind the design of ReST-MCTS*.
```
First, the main reason **we design a different reward and value for MCTS lies in the defect of conventional reward/value for the case of reasoning**. We believe that different reasoning steps (actions) should receive varied rewards according to the contribution they make, while conventional methods like Math-Shepherd often consider using sparse reward (only the final step receives reward) and probability of success as value. These methods omit the varied importance of steps and can lead to ineffective searches. On the other hand, modeling intrinsic rewards that follow the conventional RL paradigm is very difficult. Some approaches like AlphaLLM attempt to model the reward of intermediate steps in a descriptive way, but its implementation is complicated and the results are still unsatisfactory. These issues inspire us to look for alternative reward and value designs that may be more suitable for reasoning tasks, jumping out from the conventional RL paradigm.

```
The benefits of the design of ReST-MCTS*.
```
The design of the proposed ReST-MCTS$^*$ has three main benefits: **accurate evaluation of intermediate steps, balanced exploration and exploitation, and scalability**.

Firstly, we achieve a better evaluation of steps by incorporating the process reward $r_{s_k}$  (evaluates the probability of correctness of a single step) and the reasoning distance $m_k$ (evaluates contribution or importance, though in an indirect way). These factors involve information that can help assess a step more accurately, thus we believe involving these factors in the design of reward and value may improve the effectiveness of our method.

Secondly, we regard MCTS as a heuristic algorithm that balances exploration and exploitation, since it considers both visit frequency and expected reward in the UCT formula. Despite the differences in reward/value definition, we adopt the core UCT formula to achieve a balanced search in MCTS$^*$, resulting in better search outcomes.

Thirdly, our design enables easy acquisition of estimations of each node’s reasoning distance and quality value as illustrated in Section 3.2. This enables scaling up our self-training process, which is a significant advantage and practical benefit of our design. Furthermore, Table 2 proves that ReST-MCTS$^*$ is indeed effective among various tasks and LLMs, further verifying the scalability of our method.

```
Different definitions of value functions in traditional RL and our proposed MCTS*.
```
Compared to the expected total reward in traditional RL, our value function focuses on the evaluation of the current state. Even though the settings of reward/value are different in our work, the core of the algorithm can still be preserved and adopted. On one side, we want to further explore the nodes that make more progress in the right direction (higher quality value). On the other side, we can not simply discard the search directions that haven’t made much progress for some reason. As a special search algorithm, MCTS$^*$ inherits the core decision method of MCTS and adapts to the usage of weighted reward and quality value. It suffices to perform scalable data synthesis for iterative self-training of policy and value models, achieving significant improvements as our experiments show.

```
ReST-MCTS* focuses on the offline self-training paradigm.
```
It is crucial to emphasize that our methodology predominantly focuses on the offline self-training paradigm. Within this framework, data is generated through MCTS$^*$ using a static policy and critic within each iteration, as elaborated in Section 3.2. The newly synthesized data is verified and filtered according to the ground truth, rather than the outputs of the value model. Subsequently, this filtered data is utilized to individually train the policy and value models, rendering our algorithm offline and relatively stable.

---

### Public Comment · ~Yinhe_Zheng1 · 2025-01-13
**Definition of $r_{s_k}$**

In the paper, the reward from PRM is defined as $r_{s_k}=1-r_{s_k}^{HE}$. I am wondering why would you define $r_{s_k}$ in this way? This definition brought significent burden in understanding eq.2 since one would naturally expect a 'more correct' step will has a larger $r_{s_k}$, not a smaller $r_{s_k}$.

Why not defining $r_{s_k}=r_{s_k}^{HE}$?

---

### Decision · Program_Chairs · 2024-09-25

**Decision:**

Accept (poster)

**Comment:**

Despite raising several concerns regarding soundness of the weighted reward approach, clarity, positioning within existing literature, and evaluation, the authors thoroughly addressed these concerns and all reviewers recommended acceptance with varying strength. I will therefore join them in recommending acceptance, but will note that, having read the discussion, I am not satisfied beyond all doubt that the weighted reward approach is indeed sound, and would encourage the authors to justify it more thoroughly. I'd also like to note that unpublished works like AlphaLLM, which seem to have only been posted on arXiv with the footnote “Work in progress”, should not be considered prior work for peer reviewed publications, although the authors would do well to acknowledge them.